# Downregulation of Dickkopf-3, a Wnt antagonist elevated in Alzheimer's disease, restores synapse integrity and memory in a disease mouse model

Nuria Martin Flores[1†], Marina Podpolny[1†], Faye McLeod[1], Isaac Workman[1], Karen Crawford[2], Dobril Ivanov[2], Ganna Leonenko[2], Valentina Escott-Price[2,3], Patricia C Salinas[1]*

[1]Department of Cell and Developmental Biology, Division of Biosciences, University College London, London, United Kingdom; [2]Division of Psychological Medicine and Clinical Neurosciences, Cardiff University, Cardiff, United Kingdom; [3]UK Dementia Research Institute, Cardiff University, Cardiff, United Kingdom

*For correspondence:
p.salinas@ucl.ac.uk

†These authors contributed equally to this work

Competing interest: The authors declare that no competing interests exist.

**Abstract** Increasing evidence supports a role for deficient Wnt signaling in Alzheimer's disease (AD). Studies reveal that the secreted Wnt antagonist Dickkopf-3 (DKK3) colocalizes to amyloid plaques in AD patients. Here, we investigate the contribution of DKK3 to synapse integrity in healthy and AD brains. Our findings show that DKK3 expression is upregulated in the brains of AD subjects and that DKK3 protein levels increase at early stages in the disease. In hAPP-J20 and hAPP[NL-G-F/NL-G-F] mouse AD models, extracellular DKK3 levels are increased and DKK3 accumulates at dystrophic neuronal processes around plaques. Functionally, DKK3 triggers the loss of excitatory synapses through blockade of the Wnt/GSK3β signaling with a concomitant increase in inhibitory synapses via activation of the Wnt/JNK pathway. In contrast, DKK3 knockdown restores synapse number and memory in hAPP-J20 mice. Collectively, our findings identify DKK3 as a novel driver of synaptic defects and memory impairment in AD.

## eLife assessment

This **important** manuscript investigates the roles of DKK3 in AD synapse integrity. Although previous work has identified the involvement of Wnt and DKK1 in synaptic physiology, this study provides **compelling** evidence that suppression of DKK3 rescues the changes in excitatory synapse numbers, as well as memory deficits in an established AD model mice. The authors provide both gain and loss of function data that support the main conclusion and advance our understanding of the mechanisms by which Wnt pathway mediates early synaptic dysfunction in AD models.

## Introduction

Alzheimer's disease (AD) is the most common form of dementia in the aging population. The disease is characterized by progressive synaptic dysfunction and loss, early signatures that correlate with cognitive decline in AD and precede both neuronal death and the onset of severe dementia by at least 10 years (*Mucke and Selkoe, 2012*; *Selkoe and Hardy, 2016*). Current AD models suggest that amyloid-β (Aβ) initiates a pathophysiological cascade leading to synapse failure and eventually cognitive decline. Although the primary neuropathological hallmarks of AD are amyloid plaques and neurofibrillary tangles, soluble Aβ oligomers (Aβo) are considered one of the key toxic proteins

**eLife digest** Alzheimer's disease is the most common form of dementia worldwide. The cognitive decline typically observed in this condition is associated with the weakening and eventually the loss of synapses, the structures that allow neurons to communicate. Increasing evidence points to this deterioration being linked to deficiency in the Wnt signalling pathway, a cascade of molecular events crucial for brain function and development.

The DKK protein family helps to tightly regulate the Wnt pathway by dampening its activity. Previous work suggests that DKK proteins could also be connected to Alzheimer's disease. For example, an elevated amount of DKK1 leads to synapse and memory defects in mice, while brain production of DKK1 is increased in individuals with late Alzheimer's.

More recent studies show high levels of another DKK protein, DKK3, in Alzheimer's patients. This protein is also present in the harmful amyloid-β aggregates, named 'plaques', that typically form in the brain in this condition. Despite these findings, how DKK3 participates in synaptic health remains unclear.

To address this question, Martin-Flores, Podpolny et al. tracked DKK3 levels in the brains of Alzheimer's patients, revealing that they increase early in the disease. Additional experiments in Alzheimer's mouse models suggested that DKK3 secretion rise before amyloid-β plaques form, with the protein then accumulating in abnormal neuronal structures present in the surroundings of these toxic deposits.

Martin-Flores, Podpolny et al. then examined the impact of DKK3 on the Wnt pathway, and ultimately, on the balance between synapses that control neuronal activity. These experiments showed that elevated DKK3 levels are linked to a loss of synapses which are excitatory, with a concomitant increase in those that are inhibitory. Crucially, reducing DKK3 levels in a mouse model of Alzheimer's restored this synaptic balance and improved memory, highlighting DKK3 as a potential driver of cognitive impairment.

Overall, these findings help to refine our understanding of the molecular mechanisms that contribute to synaptic impairment in Alzheimer's disease. They may also be relevant for researchers studying other conditions that involve aberrant activity of the Wnt pathway, such as cancer.

driving synapse dysfunction (*Mucke and Selkoe, 2012*; *Selkoe and Hardy, 2016*; *Walsh et al., 2002*). However, the exact mechanisms by which Aβo impair synapse function and cause their degeneration are not fully understood.

Increasing evidence suggests that Wnt signaling is compromised in AD, contributing to synapse degeneration. Wnts are secreted proteins that play a crucial role in synapse formation, synaptic plasticity, and synapse integrity (*McLeod and Salinas, 2018*). The canonical Wnt pathway is particularly impaired in AD. For example, levels of the secreted Wnt antagonist Dickkopf-1 (DKK1) are increased in the brain of AD patients and AD models (*Caricasole et al., 2004*; *Purro et al., 2012*; *Rosi et al., 2010*). DKK1 promotes synapse degeneration and its blockade protects against Aβ-induced dendritic spine and synapse loss (*Marzo et al., 2016*; *Purro et al., 2012*; *Sellers et al., 2018*). Supporting the role of deficient Wnt signaling in AD, three genetic variants of *LRP6,* a crucial Wnt co-receptor, are linked to late-onset AD and confer decreased Wnt signaling in cell lines (*Alarcón et al., 2013*; *De Ferrari et al., 2007*). Notably, mice carrying the *Lrp6*-Valine variant exhibit increased synapse vulnerability during aging and in AD (*Jones et al., 2023*). Furthermore, loss-of-function of *Lrp6* exacerbates amyloid pathology in an AD mouse model (*Liu et al., 2014*). In addition, Frizzled-1 (Fz1) and Fz7, Wnt receptors present at synapses, are downregulated in the hippocampus of AD subjects and AD models (*Palomer et al., 2022*). However, the molecular mechanisms by which deficient Wnt signaling contributes to synaptic defects in AD are poorly understood. Importantly, it remains unexplored whether amelioration of Wnt deficiency restores synaptic connectivity and memory in AD.

Dickkopf-3 (DKK3), a member of the secreted Wnt antagonist DKK family, could contribute to AD pathogenesis. Like other DKKs, DKK3 has two cysteine-rich domains but it also contains an elongated N-terminus with a Soggy domain (*Krupnik et al., 1999*; *Niehrs, 2006*). Although studies suggest that DKK3 antagonizes the Wnt canonical pathway (*Caricasole et al., 2003*; *Mizobuchi et al., 2008*; *Zhu et al., 2014*), the function of DKK3 in the adult brain is unclear. *Dkk3* knock-out mice are viable

 Research article

and do not exhibit morphological alterations in the brain, but female mice manifest hyperlocomotion (*Barrantes et al., 2006*). Recent findings indicate that DKK3 is increased in plasma, and cerebrospinal fluid (CSF), and accumulates in Aβ plaques in the human AD brain (*Bruggink et al., 2015*; *Drummond et al., 2017*; *Xiong et al., 2019*). However, the impact of DKK3 on synapses and cognitive function, which are affected by deficient Wnt signalling (*Jones et al., 2023*; *Marzo et al., 2016*), in AD remains to be studied. Studies on DKK3 in AD would shed new light on the mechanisms that contribute to synapse vulnerability in AD.

Here, we investigate the role of DKK3 in the integrity and function of excitatory and inhibitory synapses in healthy and AD brains. Our RNAseq analyses reveal that *DKK3* expression is increased in the brains of AD patients. Consistently, we found that DKK3 protein is increased in the human AD brain from early stages of the disease. In two AD mouse models, extracellular DKK3 is increased in the hippocampus before substantial plaque deposition. As the pathology progresses in the mouse AD brain, DKK3 accumulates in dystrophic neurites around amyloid plaques. Functionally, our confocal and electrophysiological studies demonstrate that increased levels of DKK3 trigger the loss of excitatory synapses with a concomitant increase in inhibitory synapses in the adult mouse hippocampus through different Wnt pathways. Crucially, in vivo downregulation of DKK3 ameliorates excitatory and inhibitory synaptic defects in the hippocampus and improves memory in an AD mouse model. Together, our findings in humans and functional studies in mice identify DKK3 as a driver of synapse pathology and cognitive impairment in AD.

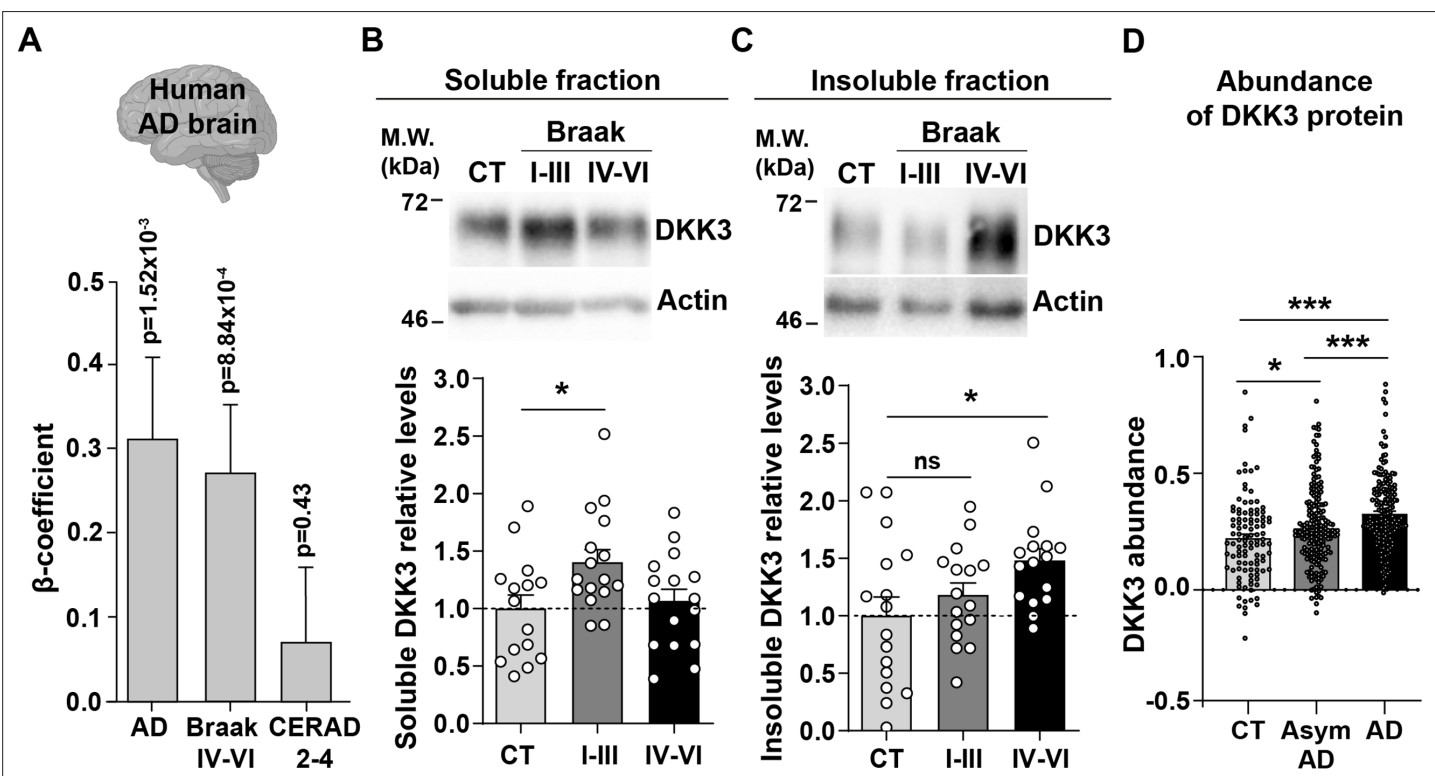

**Figure 1.** DKK3 mRNA and protein levels are increased in the human AD brain. (**A**) Temporal cortex RNAseq dataset logistic regression shows that *DKK3* mRNA levels are increased in AD cases relative to controls. Ordinal regression shows that *DKK3* is differentially expressed for Braak scores IV-VI but not for CERAD scores 2–4. (**B, C**) Representative immunoblots of DKK3 and loading control actin in (**B**) soluble and (**C**) insoluble protein fractions from the hippocampus of control (CT; n=15–16), Braak stages I-III (n=16), and Braak stage IV-VI (n=16) individuals (One-Way ANOVA test followed by Tukey's multiple comparisons). See also *Supplementary file 1*. (**D**) Abundance of DKK3 protein in dorsolateral prefrontal cortex from control CT, n=106, asymptomatic (Asym) AD (n=200), and AD (n=182) individuals was evaluated using a tandem mass tag mass spectrometry (TMT-MS) proteomic dataset study (*Johnson et al., 2022*) (ANOVA with two-sided Holm correction, AD vs CT p-value = 0.00000168, Asym AD vs. CT p-value = 0.042492481, AD vs. Asym AD = 0.000402907).

The online version of this article includes the following source data for figure 1:

**Source data 1.** Uncropped western blot gels.

# Results

## DKK3 is increased in the human AD brain

A previous study found that DKK3 is present in Aβ plaques in the brain of AD patients (**Bruggink et al., 2015**). To investigate whether DKK3 is increased in AD, we examined the expression of *DKK3* in the AD brain using RNA-seq data from the ROSMAP (**De Jager et al., 2018**), MSBB (**Wang et al., 2018**), and MayoRNAseq (**Allen et al., 2016**) datasets (n=248 controls, 379 AD cases). Logistic regression analyses revealed that *DKK3* was upregulated in AD cases (regression β-coefficient=0.31; p-value = $1.52 \times 10^{-3}$). In addition, ordinal regression analyses showed that *DKK3* was differentially expressed in relation to Braak scores, a measure of neurofibrillary tangle pathology (**Braak et al., 2006**; regression β-coefficient=0.27; p-value = $8.84 \times 10^{-4}$), but not to CERAD scores, a measure of neuritic plaque density (**Mirra et al., 1991**; regression β-coefficient: 0.07; p-value = 0.43; *Figure 1A*). These results indicate that the expression of *DKK3* is increased in the brain of human AD patients.

We next assessed DKK3 protein levels in the hippocampus of AD patients at different disease stages based on their Braak status. We evaluated healthy individuals and patients with Braak stages I-III and IV-VI (n=16 per group, *Supplementary file 1*). Given that DKK3 is found at Aβ plaques in the human AD brain (**Bruggink et al., 2015**; **Drummond et al., 2017**), we analyzed DKK3 protein in the soluble and insoluble fractions. Soluble DKK3 levels were increased in Braak I-III patients when compared to control subjects (1.4-fold increase), but no changes were observed in Braak IV-VI (*Figure 1B*). In the insoluble fraction, in contrast, DKK3 protein levels were increased in Braak IV-VI (1.51-fold increase; *Figure 1C*), which could be consistent with the presence of DKK3 in Aβ plaques. To explore if DKK3 increases at early stages of AD, we assessed DKK3 abundance in brain tissue from the dorsolateral prefrontal cortex using a published proteomic study comprised of 106 controls, 182 AD cases, and 200 asymptomatic AD cases (**Johnson et al., 2022**). The asymptomatic cases are at an early preclinical stage of AD in which patients exhibit Aβ and tau pathology but no significant cognitive impairment (**Jack et al., 2018**; **Johnson et al., 2022**). In line with our above results, DKK3 protein was elevated in AD cases. Importantly, DKK3 was increased in asymptomatic AD cases and further increased in AD cases when compared to asymptomatic AD (*Figure 1D*). Together, these results demonstrate that DKK3 mRNA and DKK3 protein levels are elevated in the brains of AD patients and increased with the progression of AD. Moreover, DKK3 protein is re-distributed from soluble to insoluble fractions with disease progression.

## DKK3 accumulates at atrophic neurites around amyloid plaques in AD mouse models

We next analyzed where DKK3 protein is present in healthy and AD mouse brains. *Dkk3* is expressed in excitatory neurons in several brain areas including the hippocampus and neocortex (**Barrantes et al., 2006**; **Meister et al., 2015**; **Thompson et al., 2008**). To study the distribution of DKK3 protein, we used a specific DKK3 antibody, validated with brain samples from total knockout *Dkk3*⁻/⁻ mice by western blot (*Figure 2—figure supplement 1A*) and confocal microscopy (*Figure 2—figure supplement 1B*). In the hippocampus, DKK3 protein was highly abundant in neurons of the CA1, CA2, and CA3 pyramidal layers but not in the dentate gyrus (DG) granule cell layer (*Figure 2—figure supplement 1C*), consistent with previous works studying *Dkk3* expression (**Barrantes et al., 2006**; **Thompson et al., 2008**). In addition, DKK3 protein was present at lower levels in a subset of astrocytes (GFAP-positive cells; *Figure 2—figure supplement 1D*), but not in microglia (IBA1-positive cells; *Figure 2—figure supplement 1E*). Thus, principal neurons, followed by astrocytes, are the main source of DKK3 protein in the adult mouse hippocampus.

To further understand the role of DKK3 in the AD brain, we evaluated its localization within the hippocampus of two AD mouse models when Aβ plaques are present. Aβ plaques are complex structures closely associated with atrophic axons and dendrites of nearby neurons and with glial cells (*Figure 2A*). Using confocal microscopy, we investigated the presence of DKK3 in hippocampal Aβ plaques using anti-Aβ (6E10) in 18-month-old J20 (**Mucke et al., 2000a**) and 8-month-old hAPP^NL-G-F/NL-G-F (NLGF) mice (**Saito et al., 2014**). DKK3 was present at Aβ plaques in both AD lines (*Figure 2B and Figure 2—figure supplement 2*). Furthermore, DKK3 was present in both diffuse Aβ plaques (6E10-positive but ThioS negative) and dense-core plaques (positive for 6E10 and ThioS; *Figure 2—figure supplement 2B*). Importantly, 70% of dense-core Aβ plaques contained DKK3 (*Figure 2C*). This localization was specific for DKK3, as other secreted proteins such as Wnt7a/b did not localize

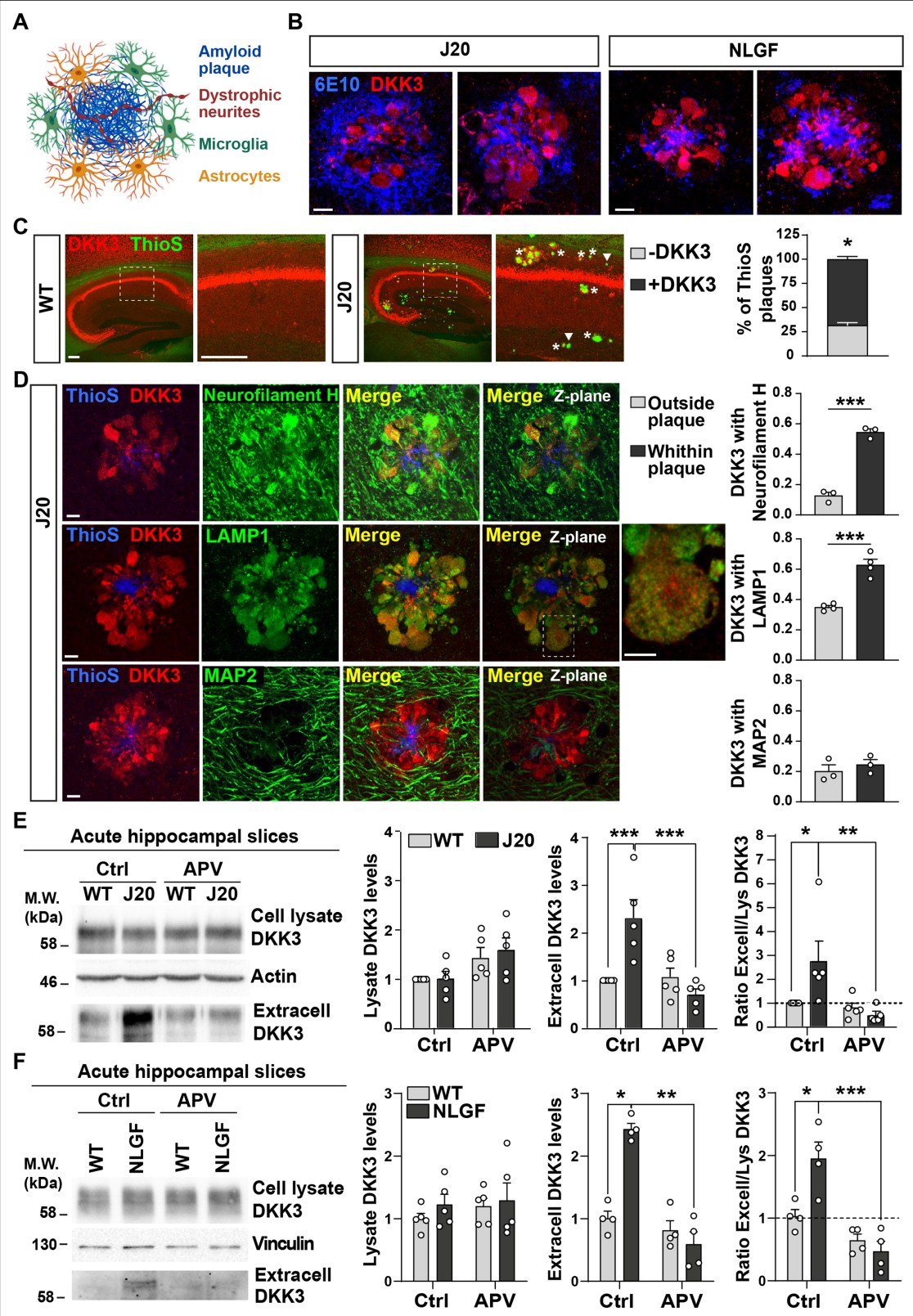

**Figure 2.** DKK3 localizes to dystrophic neurites around Aβ plaques, and DKK3 extracellular levels are increased in the brain of AD mouse models. (**A**) Diagram of the components of an Aβ plaque (blue), astrocytes (orange), microglia (green), and dystrophic neurites (red). (**B**) Confocal images of DKK3 protein (red) and amyloid plaques stained with the 6E10 antibody (blue) in the hippocampus of 18-month-old J20 and 8 months NLGF mice. Scale bar = 10 μm. (**C**) Confocal images of DKK3 (red) and Aβ plaques labeled by Thioflavin S (ThioS; green) in the hippocampus of 18-month-old WT

*Figure 2 continued on next page*

*Figure 2 continued*

and J20 mice. ThioS +plaques not containing DKK3 (- DKK3; arrowheads), ThioS +plaques containing DKK3 (+DKK3; asterisks). Scale bar = 150 μm and 100 μm in zoom-in pictures. Graph depicts quantification of the percentage of ThioS +plaques containing or not DKK3 (Student's T-test, n=3 animals per genotype). (D) Z-stack confocal images show that DKK3 (red) accumulates at Aβ plaques (ThioS; blue) and colocalizes with atrophic axons (Neurofilament-H; green and LAMP1; green) but not with dendrites (MAP2; green). XY views of one plane are shown in the last panel. For LAMP1, a zoom-in picture showing colocalization between DKK3 and LAMP1 puncta is shown. Scale bar = 6 μm. Graphs show Pearson's correlation coefficient between DKK3 and Neurofilament-H, LAMP1, or MAP2, n=3–4 animals. (E, F) Immunoblot images show DKK3 levels in the cell lysate and secreted fraction of acute hippocampal slices of (E) 3–4 month-old WT and J20 mice or (F) 2–3 months old WT and NLGF mice. Slices were incubated with vehicle (Ctrl) or APV for 3 hr. Actin or Vinculin was used as a loading control in the homogenate. Graphs show densitometric quantifications of lysate and extracellular (extracell) DKK3 levels relative to control and the ratio of extracellular/lysate DKK3 levels (Two-Way ANOVA followed by Tukey's post-hoc test; n=4–5 animals).

The online version of this article includes the following source data and figure supplement(s) for figure 2:

**Source data 1.** Uncropped western blot gels.

**Figure supplement 1.** DKK3 is present in principal neurons of the mouse hippocampus.

**Figure supplement 1—source data 1.** Uncropped western blot gels.

**Figure supplement 2.** DKK3 accumulates in Aβ plaques in the hippocampus of J20 and NLGF mice and DKK3 levels are increased by Aβ.

**Figure supplement 2—source data 1.** Uncropped western blot gels.

**Figure supplement 3.** cLTD, but not cLTP, increases the levels of extracellular DKK3.

**Figure supplement 3—source data 1.** Uncropped western blot gels.

to Aβ plaques (*Figure 2—figure supplement 2C*). Furthermore, DKK3 was absent from astrocytes and microglia in Aβ plaques but specifically colocalized with Neurofilament-H +dystrophic neurites (*Figure 2D* and *Figure 2—figure supplement 2D*). In addition, DKK3 also colocalized with LAMP1 in dystrophic neurites, which are visualized as axonal spheroids (*Figure 2D*). DAPI staining further revealed that the deposition of DKK3 within amyloid plaques was not in cell body inclusions (*Figure 2—figure supplement 2E*). Together, these results indicate that DKK3 is present in atrophic neurites around amyloid plaques.

## Extracellular levels of DKK3 increase in the AD mouse brain through NMDAR activation

The accumulation of DKK3 in Aβ plaques in AD (*Figure 2*) and our finding that DKK3 protein is elevated in the brain of AD patients (*Figure 1C and D*) led us to investigate whether DKK3 levels are altered in the mouse AD brain. Given that DKK3 is a secreted protein, we examined total and extracellular levels of DKK3 from acute hippocampal slices of wild-type (WT) and J20 mice at 3–4 months before plaques appear. Although the total DKK3 levels did not differ between J20 and WT mice in the brain homogenate, DKK3 levels were significantly increased (2.54-fold) in the extracellular fraction of J20 mice (*Figure 2E*). In a second AD mouse model, NLGF, extracellular DKK3 levels were also elevated by 2.43-fold in brain slices of these animals at 2–3 months (*Figure 2F*). Importantly, the ratio of extracellular to total DKK3 levels was significantly higher in J20 and NLGF when compared to their respective controls (*Figure 2E and F*), suggesting that DKK3 secretion is enhanced in AD mouse brains.

Our next studies focused on understanding the mechanisms underlying the extracellular increase of DKK3 in AD mice. Mounting evidence demonstrates that Aβo trigger the overactivation of *N*-methyl-D-aspartate (NMDA) receptors (NMDARs), which contributes to long-term depression (LTD) in AD models (*Li et al., 2011*; *Mucke and Selkoe, 2012*). We therefore investigated this question by using (2 R)-amino-5-phosphonovaleric acid (APV) to block NMDARs. APV completely prevented the increase of DKK3 in the extracellular fraction of J20 and NLGF brain slices (*Figure 2E and F*). Given the increased levels of Aβ in these two mouse models (*Mucke et al., 2000a*; *Saito et al., 2014*), we evaluated whether Aβ increases DKK3 levels. We treated hippocampal neurons with Aβo (Aβ$_{1-42}$) or the reverse Aβ$_{42-1}$ control peptide. Aβo increased DKK3 protein levels by 2.50- and 2.48-fold in the cellular lysate and the extracellular fraction respectively (*Figure 2—figure supplement 2F–H*), indicating that Aβo increased the overall levels of DKK3 in hippocampal neurons. Treatment of neurons with APV in the presence of Aβo decreased extracellular DKK3 levels to 1.48-fold. Although this reduction did not reach statistical significance using a Kruskal-Wallis with Dunn's test (p=0.0726), it

was statistically significant using a t-test (p=0.0384). These results suggest that blockade of NMDARs partially occludes the ability of Aβo to increase DKK3 levels in the extracellular fraction.

Next, we examined whether DKK3 levels were regulated by NMDAR-mediated synaptic plasticity by performing glycine-induced chemical long-term potentiation (cLTP) or NMDA-induced cLTD in cultured hippocampal neurons. Induction of cLTP did not affect DKK3 protein levels in the cellular or extracellular fractions (*Figure 2—figure supplement 3A*). In contrast, cLTD significantly increased the levels of extracellular DKK3 without affecting the levels in the cellular fractions (*Figure 2—figure supplement 3B*). Similar results were obtained using brain slices after cLTD-induction (*Figure 2—figure supplement 3C*). To test if the increase in extracellular DKK3 was due to changes in vesicular trafficking of DKK3, we used brefeldin A (BFA), which interrupts vesicle trafficking and exocytosis (*Brewer et al., 2022*; *He et al., 2015*; *Katsinelos et al., 2018*). We found that BFA treatment significantly reduced DKK3 levels in the extracellular space under control conditions and completely prevented the increase in DKK3 levels following cLTD-induction (*Figure 2—figure supplement 3C*). The lack of a difference in DKK3 levels in the total homogenate could be explained by the fact that only a small fraction of cellular DKK3 is released into the extracellular media, as supported by our findings that DKK3 was less abundant in the extracellular fraction when compared to the total homogenate (*Figure 2—figure supplement 3D*). Together, these results strongly suggest that trafficking/secretion of DKK3 was enhanced by NMDAR-mediated cLTD-induction.

## DKK3 differentially affects excitatory and inhibitory synapses in the adult hippocampus

Given that DKK1, a member of the Dkk family, leads to excitatory synapse disassembly and synaptic plasticity defects (*Galli et al., 2021*; *Galli et al., 2014*; *Marzo et al., 2016*), we evaluated the impact of increased DKK3 levels on synapses by performing ex vivo gain-of-function experiments using brain slices (*Figure 3A*). We focused on the CA3 region as DKK3 is highly expressed in this region (*Thompson et al., 2008*; *Figure 2—figure supplement 1C*), and is required for encoding spatial and other episodic memories, processes which are impaired in AD (*Deuker et al., 2014*). Moreover, we previously reported that Aβo trigger synapse loss in the CA3 *stratum radiatum* (SR) region of the hippocampus (*Purro et al., 2012*). Gain-of-function of DKK3 reduced the puncta number of the excitatory presynaptic marker vGLUT1 (by 38.99%), the postsynaptic marker PSD-95 (by 32.58%), and the total number of excitatory synapses (by 60.85%), determined by the colocalization of these synaptic markers, in the CA3 SR (*Figure 3B*). These synaptic changes were not due to neuronal death (*Figure 3—figure supplement 1A and B*). Patch-clamp recordings of CA3 neurons revealed that DKK3 gain-of-function decreased the frequency of miniature excitatory postsynaptic currents (mEPSC) by 48.54% but did not affect their amplitude (*Figure 3C*).

We further investigated whether DKK3 gain-of-function affects inhibitory synapses. DKK3 increased the density of puncta for the inhibitory postsynaptic marker gephyrin (by 55.39%) without affecting the density of the inhibitory presynaptic marker vGAT in the CA3 SR (*Figure 3D*). Notably, DKK3 increased the number of inhibitory synapses (by 65.99%; *Figure 3D*) based on the colocalization of the pre and postsynaptic markers. Patch-clamp recordings of CA3 neurons uncovered that DKK3 increased the frequency of miniature inhibitory postsynaptic currents (mIPSC) by 65.56% but did not affect their amplitude (*Figure 3E*). In the CA1 SR, DKK3 induced similar effects on excitatory and inhibitory synapse density (*Figure 3—figure supplement 1C and C*). Together, these results demonstrate that DKK3 gain-of-function decreases excitatory synapse number but increases inhibitory synapse number in the adult hippocampus.

## DKK3 regulates excitatory synapse number through the Wnt/GSK3β pathway and inhibitory synapse number through Wnt/JNK cascade

We next examined the Wnt signaling pathways mediating DKK3-induced synaptic changes. Wnts can signal through different pathways, including the Wnt/GSK3 and Wnt/JNK cascades (*Niehrs, 2012*; *Nusse and Clevers, 2017*; *Figure 4A&D*). A previous study showed that DKK1 induces synapse loss by blocking the canonical Wnt pathway in the hippocampus (*Marzo et al., 2016*). To investigate if DKK3 triggers synaptic changes through this pathway, we evaluated the puncta density of β-catenin, which is degraded upon inhibition of Wnt/GSK3β signaling (*Nusse and Clevers, 2017*). To exclude changes in β-catenin density due to synapse loss, we measured the density of extra-synaptic β-catenin

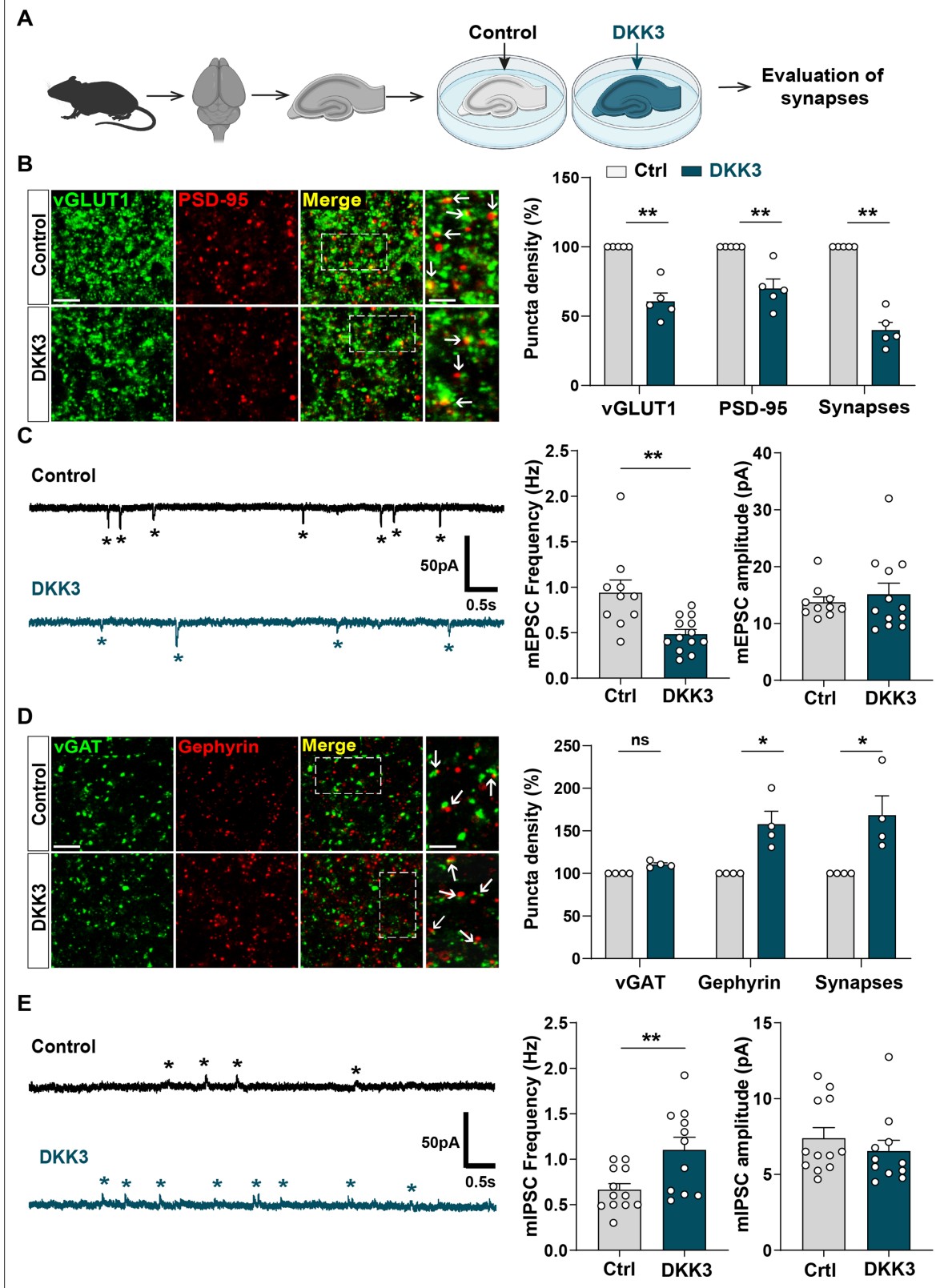

**Figure 3.** Gain-of-function of DKK3 leads to opposing effects on the number of excitatory and inhibitory synapses in the hippocampus. (**A**) Diagram depicting the treatment of hippocampal brain slices obtained from 3-month-old adult WT mice with vehicle (Ctrl) or recombinant DKK3 protein. Synapses were evaluated by confocal microscopy and electrophysiological recordings. (**B**) Confocal images of the CA3 SR region labeled with the presynaptic excitatory marker vGLUT1 (green) and the postsynaptic marker PSD-95 (red). Arrows indicate excitatory synapses as colocalized pre- and

*Figure 3 continued on next page*

*Figure 3 continued*

postsynaptic puncta. Scale bar = 5 µm and 2.5 µm in zoomed-in pictures. Quantification is shown on the right-hand side (Mann-Whitney test, n=5 animals per condition). (**C**) Representative mEPSC traces recorded at –60 mV from CA3 cells. Stars indicate mEPSC events. Quantification of mEPSC frequency and amplitude is shown on the right-hand side (Mann-Whitney test, n=10–13 cells from five animals). (**D**) Confocal images of the CA3 SR region labeled with the presynaptic inhibitory marker vGAT (green) and the postsynaptic marker gephyrin (red). Arrows indicate inhibitory synapses as colocalized pre- and postsynaptic puncta. Scale bar = 5 µm and 2.5 µm in zoomed-in pictures. Quantification is shown on the right-hand side (Mann-Whitney test, n=4 animals per condition). (**E**) Representative mIPSC traces recorded at 0 mV from CA3 cells. Stars indicate mIPSC events. Quantification of mIPSC frequency and amplitude is shown on the right-hand side (Student's T-test for mIPSC frequency and Mann-Whitney test for mIPSC amplitude, n=11–12 cells from five to seven animals).

The online version of this article includes the following figure supplement(s) for figure 3:

**Figure supplement 1.** DKK3 triggers changes in excitatory and inhibitory synapses in the absence of cell death.

as we had done before (*Galli et al., 2014*). Indeed, DKK3 decreased the number of extra-synaptic β-catenin puncta in hippocampal slices, which was restored when the Wnt/GSK3β cascade was activated using the GSK3 inhibitor 6-bromoindirubin-3'-oxime (BIO) (*Marzo et al., 2016*; *Figure 4B*). Importantly, BIO increased the density of extra-synaptic β-catenin puncta under control conditions, confirming the activation of canonical Wnt signaling (*Figure 4B*). We next explored whether activation of canonical Wnt signaling prevented DKK3-induced synaptic changes. At the concentration and time used, BIO did not affect the number of excitatory synapses under control brain slices, but completely blocked the DKK3-induced loss of excitatory synapses (*Figure 4C*). Similar results were obtained with CHIR99021, another highly specific GSK3 inhibitor (*Ring et al., 2003*; *Figure 4—figure supplement 1A*). Thus, DKK3 affects excitatory synapse number through the Wnt/GSK3β pathway.

Next, we investigated if DKK3 increases inhibitory synapses through canonical Wnt signaling by blocking GSK3 using BIO. In contrast to excitatory synapses, BIO did not affect DKK3's impact on inhibitory synapse density (*Figure 4—figure supplement 1B*). Therefore, DKK3 regulates inhibitory synapse density independently of the Wnt/GSK3β pathway. Previous studies showed that DKK1 concomitantly inhibits the Wnt/GSK3β pathway and activates the Wnt/Planar Cell Polarity (PCP) signaling cascade (*Caneparo et al., 2007*; *Killick et al., 2014*; *Marzo et al., 2016*). The PCP pathway activates c-Jun N-terminal kinase (JNK) (*Figure 4D*), which has been implicated in Aβ toxicity (*Killick et al., 2014*). In brain slices, DKK3 increased phospho-JNK, a readout for the JNK activation (*Figure 4E*). This increase was blocked by the JNK inhibitor CC-930 (*Plantevin Krenitsky et al., 2012*; *Figure 4E*), indicating that DKK3 activates the Wnt/JNK signaling pathway. We next tested the effect of CC-930 on inhibitory synapses and found that this JNK inhibitor blocked the DKK3-induced increase in the number of gephyrin puncta and inhibitory synapse density (*Figure 4F*). In contrast, JNK inhibition did not block the effect of DKK3 on excitatory synapses (*Figure 4—figure supplement 1C*). Together, our results indicate that DKK3 induces the loss of excitatory synapses through inhibition of Wnt/GSK3β signaling but increases inhibitory synapses through activation of the Wnt/JNK pathway.

## In vivo DKK3 loss-of-function decreases inhibitory synapses without affecting excitatory synapses in the healthy adult brain

We next studied the in vivo role of DKK3 by downregulating DKK3 in adult WT mice using a viral transduction approach. DKK3 was knocked down in the CA3 region of the hippocampus using AAV9 expressing enhanced green fluorescent protein (EGFP) and scramble shRNA (Scr shRNA) or shRNA against *Dkk3* (*Dkk3* shRNA; *Figure 5A*), and synapses were evaluated a month later. This approach led to approximately 85% knockdown of DKK3 at the injection site (*Figure 5A*). Excitatory and inhibitory synapses were assessed by confocal microscopy and by whole-cell patch-clamp recordings. In contrast to gain-of-function experiments, DKK3 silencing did not affect excitatory synapse number (*Figure 5B*) or mEPSC frequency (*Figure 5C*). However, DKK3 loss-of-function decreased the amplitude of mEPSCs (by 28.18%, *Figure 5C*). Conversely, knockdown of DKK3 reduced the number of inhibitory synapses (by 37.30%, *Figure 5D*), the frequency of mIPSCs (by 71.76%), and their amplitude (by 35.02%, *Figure 5E*). Thus, downregulation of DKK3 in WT brain reduces the number of inhibitory synapses without affecting excitatory synapses, suggesting that endogenous DKK3 is required for the maintenance of inhibitory synapses but not for the integrity of excitatory synapses in the healthy adult brain.

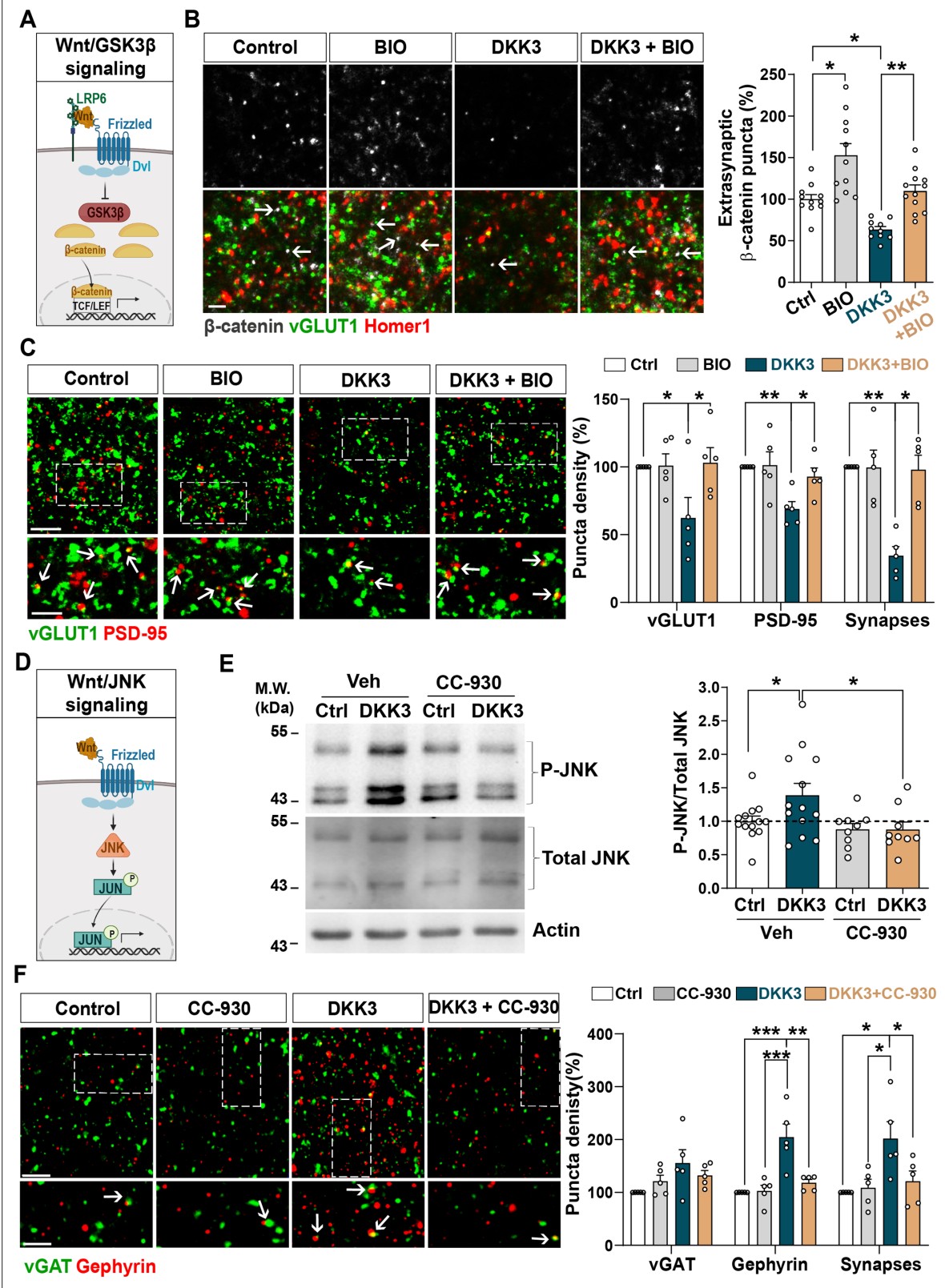

**Figure 4.** DKK3 regulates excitatory and inhibitory synapse number through the Wnt/GSK3β and Wnt/JNK pathways respectively. (**A**) Diagram of the canonical Wnt pathway through inhibition of GSK3β (Wnt/GSK3β pathway), resulting in elevation of β-catenin and transcriptional activation via TCF/LEF. (**B**) Confocal images show excitatory synapses, visualized by colocalization of vGLUT1 (green) and Homer1 (red), as well as β-catenin puncta (grey) in the CA3 SR after treatment with vehicle (Ctrl) or DKK3 in the absence or presence of BIO. Arrows indicate extra-synaptic β-catenin puncta. Scale bar =

*Figure 4 continued on next page*

*Figure 4 continued*

5 µm. Quantification of extrasynaptic β-catenin puncta density as a percentage of control is shown on the right-hand side (Two-Way ANOVA followed by Tukey's multiple comparisons, n=2–3 brain slices/animal from five animals). (**C**) Confocal images show excitatory synapses (co-localized vGLUT1 puncta in green and PSD-95 puncta in red) in the CA3 SR after treatment with vehicle (Ctrl) or DKK3 in the absence or presence of BIO. Scale bar = 5 µm and 2.5 µm. Graph shows the quantification of puncta density of pre- and postsynaptic markers and excitatory synapses as a percentage of control (Kruskal-Wallis followed by Dunn's multiple comparisons, n=5 animals). (**D**) Diagram of the Wnt pathway through activation of JNK (Wnt/JNK pathway), resulting in increased levels of phospho-JNK and transcriptional changes. (**E**) Representative immunoblots of phospho-JNK Thr183/Tyr185 (P-JNK) and total JNK of brain slices treated with DKK3 and/or the JNK inhibitor CC-930. Actin was used as a loading control. Graph shows densitometric quantification of P-JNK vs. total JNK relative to the control condition (Kruskal-Wallis followed by Dunn's multiple comparisons, n=2 brain slices/animal from four to five animals). (**F**) Confocal images showing inhibitory synapses defined by the colocalization of vGAT (green) and gephyrin (red) puncta in the CA3 SR after treatment with vehicle (Ctrl) or DKK3 in the absence or presence of CC-930. Scale bar = 5 µm and 2.5 µm. Graph shows the quantification of puncta density of pre and postsynaptic markers and inhibitory synapses as a percentage of control (Kruskal-Wallis followed by Dunn's multiple comparisons, n=5 animals).

The online version of this article includes the following source data and figure supplement(s) for figure 4:

**Source data 1.** Uncropped western blot gels.

**Figure supplement 1.** Activation of the Wnt/GSK3β pathway blocks DKK3-induced excitatory synapse loss whereas inhibition of the Wnt/JNK pathway blocks the effect of DKK3 on inhibitory synapses.

## In vivo DKK3 loss-of-function ameliorates excitatory and inhibitory synapse changes in J20 mice

To investigate the contribution of DKK3 to synaptic changes in AD, we knocked down DKK3 in the hippocampus of J20 mice at two different disease stages using AAV9-Scr shRNA or AAV9-*Dkk3* shRNA (*Figure 6A* and *Figure 6—figure supplement 1A*). J20 mice exhibit excitatory synapse loss in the hippocampus at 4 months of age (early stage), whereas plaque deposition starts around 5 months and is widely distributed in the cortex and hippocampus by 9 months (late stage) (*Hong et al., 2016*; *Meilandt et al., 2009*; *Mucke et al., 2000a*; *Mucke et al., 2000b*). We first evaluated the impact of DKK3 knockdown on excitatory and inhibitory synapses in J20 mice at early stages. We found that 4-month-old J20 mice exhibited a 40–45% loss of excitatory synapses in the CA3 SR when compared to WT (*Figure 6B* and *Figure 6—figure supplement 1*) as previously reported (*Hong et al., 2016*). Remarkably, DKK3 knockdown restored excitatory synapse number in J20 mice (*Figure 6B* and *Figure 6—figure supplement 1*). In contrast to excitatory synapses, inhibitory synapses were increased by 20.60% in J20 mice compared to WT (*Figure 6C* and *Figure 6—figure supplement 1C*). Importantly, DKK3 silencing decreased inhibitory synapse number in these mice (*Figure 6C* and *Figure 6—figure supplement 1*). Thus, in vivo DKK3 loss-of-function ameliorates synaptic defects in J20 mice, supporting the hypothesis that DKK3 is a key contributor to synaptic changes in this AD mouse model.

A key feature of AD brains is the loss of synapses around Aβ plaques (*Koffie et al., 2012*). Therefore, we investigated whether DKK3 affects synapse number around plaques in 9-month-old J20 mice (*Figure 6A*). We observed a significant effect of distance on the density of excitatory synapses from the core of the plaque (F (6, 318)=27.26, p-value <0.0001) and inhibitory synapses (F (6, 276)=23.51, p-value <0.0001) (*Figure 6D and E*). Importantly, DKK3 silencing significantly increased the number of excitatory synapses (*Figure 6D* and *Figure 6—figure supplement 1D*) but decreased the density of inhibitory synapses around plaques (*Figure 6E* and *Figure 6—figure supplement 1E*) when compared to J20 mice injected with Scr shRNA. Importantly, knockdown of DKK3 did not affect the number or size of plaques in the CA3 SR (*Figure 6—figure supplement 1F*).

Given the role of DKK1 in synaptic changes and in AD (*Caricasole et al., 2004*; *Marzo et al., 2016*; *Purro et al., 2012*), we next investigated whether modulation of DKK3 levels affected *Dkk1* mRNA levels in the hippocampus of WT and J20 mice injected with Scr or *Dkk3* shRNA. However, no changes in *Dkk1* mRNA levels were observed (*Figure 6—figure supplement 1G*). We next investigated whether increased *Dkk1* led to changes in *Dkk3* expression. For this, we used a transgenic mouse model that expresses *Dkk1* upon induction (*iDkk1* mice; *Galli et al., 2021*; *Galli et al., 2014*; *Marzo et al., 2016*). After 14 days of *Dkk1* expression and when synaptic changes are observed in these mice, we found that *Dkk3* levels were unaltered in the hippocampus (*Figure 6—figure supplement 1H*). Thus, *Dkk3*'s expression is unaffected by *Dkk1* and vice versa. Together, these results demonstrate that loss-of-function of DKK3 ameliorates excitatory and inhibitory synapse changes in

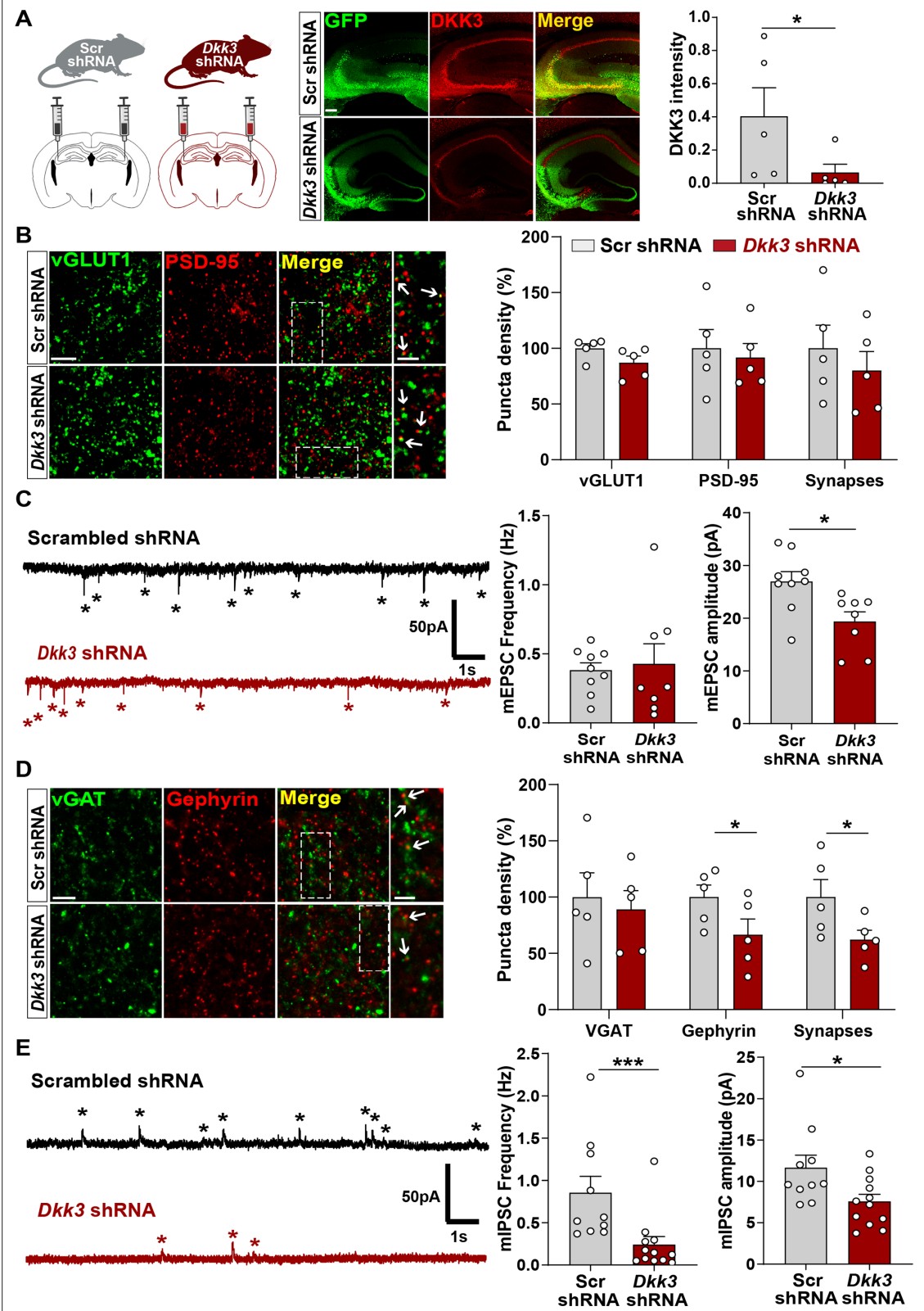

**Figure 5.** In vivo loss-of-function of DKK3 decreases inhibitory synapses but does not affect excitatory synapses in the wild-type hippocampus. (**A**) Diagram showing the experimental design. Three-month-old WT mice were injected with AAV9 scrambled (Scr) or *Dkk3* shRNA in the CA3 region. Confocal images showing GFP (green) and DKK3 (red) in Scr- and *Dkk3*-shRNA injected hippocampus. Scale bar = 145 μm. Graph shows quantification of DKK3 intensity in the area injected with the viruses. (**B**) Confocal images from CA3 SR show excitatory synapses (colocalized vGLUT1 puncta in green

*Figure 5 continued on next page*

*Figure 5 continued*

and PSD-95 puncta in red). Arrows indicate excitatory synapses. Scale bar = 5 µm and 2.5 µm in zoomed-in images. Quantification is shown on the right-hand side (Student's T-test, n=5 animals per condition). (**C**) Representative mEPSC traces recorded at –60 mV from CA3 cells. Stars indicate mEPSC events. Quantification of mEPSC frequency and amplitude is shown on the right-hand side (Student's T-test, n=8–9 cells from four animals). (**D**) Confocal images from CA3 SR show inhibitory synapses (colocalized vGAT in green and gephyrin in red). Arrows point to inhibitory synapses. Scale bar = 5 µm and 2.5 µm in zoomed-in pictures. Quantification is shown on the right-hand side (Student's T-test, n=5 animals). (**E**) Representative mIPSC traces recorded at 0 mV from CA3 cells. Stars indicate mIPSC events. Quantification of mIPSC frequency and amplitude is shown on the right-hand side (Mann-Whitney test, n=10–12 cells from six animals).

J20 mice independently of DKK1 before plaque burden starts (4-months-old), and later when amyloid plaque pathology is evident (9-month-old).

## In vivo DKK3 loss-of-function improves memory in J20 mice

The finding that DKK3 loss-of-function reverses synaptic changes at early and late stages in J20 mice led us to test whether silencing DKK3 restores hippocampal-mediated learning and memory in these mice (*Figure 7A*). Knockdown of DKK3 using viral injections did not affect exploratory activity or anxiety in J20 mice (*Figure 7—figure supplement 1A and B*). In contrast, DKK3 downregulation significantly improved spatial memory in J20 mice as evaluated by the novel object location test (NOLT) (*Figure 7B*). Next, we examined long-term spatial working memory using the Morris Water Maze (MWM). No deficiencies in vision or locomotion were observed as escape latencies did not differ between groups when the platform was visible (*Figure 7—figure supplement 1C*). We then assessed reference spatial learning using the hidden platform version of the MWM (*Figure 7C*). Performance improved significantly in all 4 groups during training, although the escape latency in J20-Scr shRNA mice remained significantly higher than that of WT-Scr shRNA mice (*Figure 7C*). Importantly, silencing DKK3 in J20 mice fully rescued this defect (*Figure 7C*). To test spatial memory, probe trials were performed on day 5 (early probe) and day 8 (late probe). In the first early probe test, J20-Scr shRNA animals traveled significantly less in the target quadrant than WT-Scr shRNA mice (*Figure 7D*). After further training, in the late probe, the time to first entrance to the target location (platform) and the distance traveled in the target quadrant were restored in the J20-*Dkk3* shRNA mice when compared to J20-Scr shRNA mice (*Figure 7E*). Together, these results demonstrate that DKK3 downregulation in the hippocampus restores cognitive function in J20 AD mice. In summary, our functional studies in mice together with our results obtained from human AD patients strongly support a role for DKK3 in synapse dysfunction and memory impairment in AD.

## Discussion

Synapse loss is the strongest correlate with cognitive impairment in AD (*Mucke and Selkoe, 2012*; *Selkoe and Hardy, 2016*). However, the mechanisms that trigger synaptic changes remain poorly understood. In this work, we investigated the function of the Wnt antagonist DKK3 on synaptic integrity and memory in the healthy and AD brain. Our functional analyses in AD models and our studies in human samples strongly support the notion that DKK3 contributes to synapse defects and memory impairment in AD.

Our analyses of brain samples from AD patients show an upregulation of DKK3 at the mRNA and protein levels. Importantly, DKK3 elevation starts from early stages as we observed increased protein levels in Braak I-III subjects and in asymptomatic cases using a published proteomic dataset (*Johnson et al., 2022*). These findings are in agreement with other proteomic studies showing increased levels of DKK3 in different brain areas, including the hippocampus and in cortical synaptosomes of AD patients (*Hesse et al., 2019*; *Xu et al., 2019*). Together, our findings in AD patients suggest that increased DKK3 levels in the brain could underlie synapse dysfunction in AD.

Amyloid plaques are a prominent neuropathological feature of AD. A previous study revealed that DKK3 is present at Aβ plaques in the brains of AD subjects (*Bruggink et al., 2015*), which was later confirmed by proteomic studies in human and mouse brains (*Drummond et al., 2017*; *Xiong et al., 2019*). Consistent with these findings, we demonstrate that DKK3 accumulates at both diffuse and dense-core Aβ plaques in two AD mouse models: the NLGF and J20 lines. Moreover, our analyses revealed that DKK3 is specifically localized at axonal spheroids. DKK3 colocalized with LAMP1,

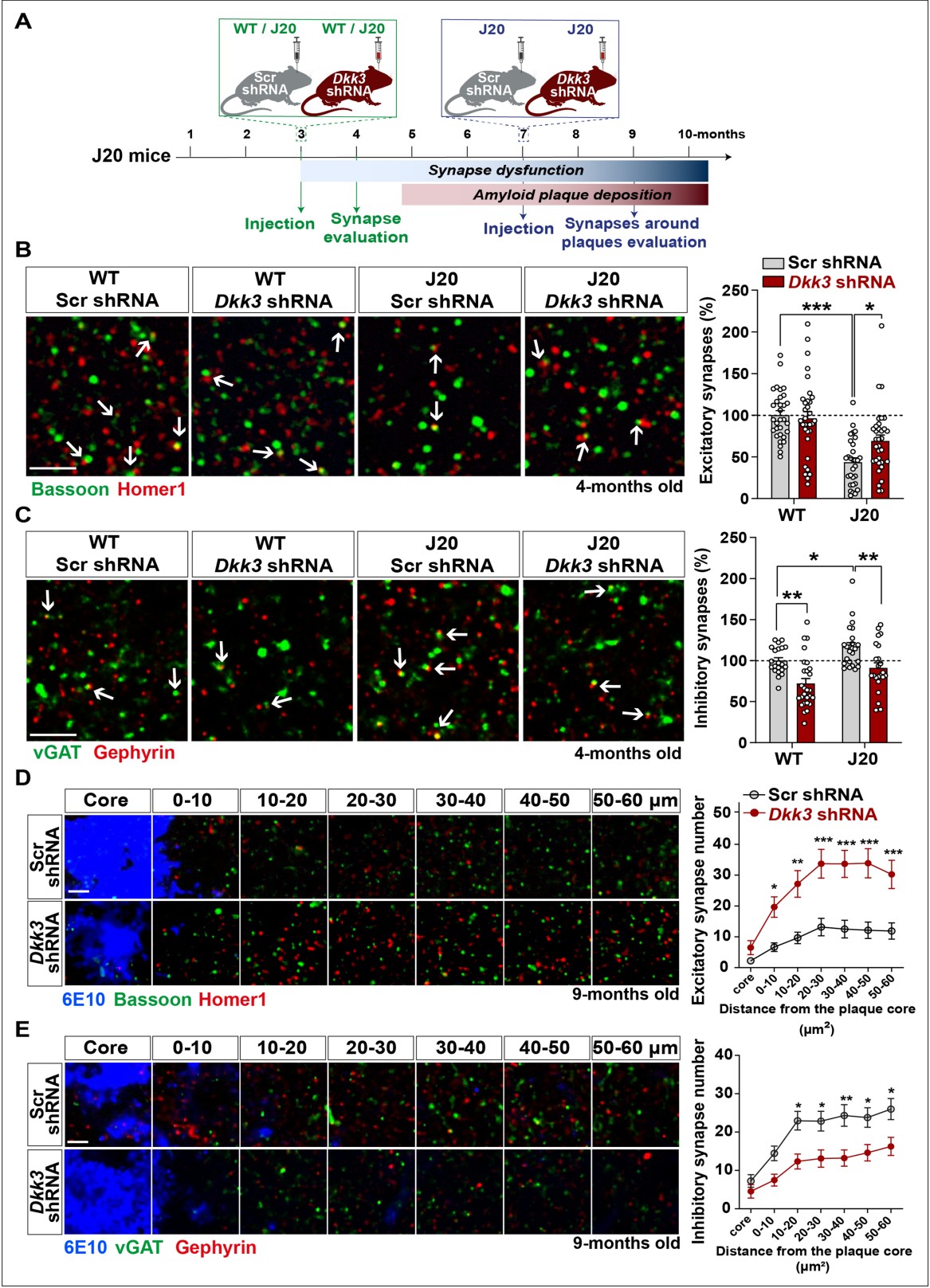

**Figure 6.** In vivo loss-of-function of DKK3 ameliorates synaptic changes in the hippocampus of J20 mice before and after Aβ plaque formation. (**A**) Diagram depicting the experimental design. In green, 3-month-old WT and J20 mice were injected bilaterally with AAV9-Scr shRNA or AAV9-*Dkk3* shRNA in the CA3 region. The density of synapses was evaluated at 4-month-old before plaque deposition starts. In blue, 7-month-old J20 mice were injected bilaterally with AAV9-Scr shRNA or AAV9-*Dkk3* shRNA in the CA3 region. The density of synapses around plaques was evaluated at 9-month-

*Figure 6 continued on next page*

*Figure 6 continued*

old. (**B, C**) Representative confocal images from the CA3 SR region of 4-month-old WT and J20 mice. Images show (**B**) excitatory synapses (Bassoon in green and Homer1 in red) and (**C**) inhibitory synapses (vGAT in green and Gephyrin in red). Arrows point to synapses. Scale bar = 2.5 μm. Quantification of synapse number as a percentage relative to WT-Scr shRNA animals is shown on the right-hand side (Two-Way ANOVA followed by Tukey's post-hoc test, n=9–11 animals per condition and 2–3 brain slices per animal). (**D, E**) Representative confocal images from the CA3 SR region of 9-month-old J20 mice. Images show an Aβ plaque (6E10; blue) and (**D**) excitatory synapses or (**C**) inhibitory synapses at different distances relative to the core of the plaque. Scale bar = 2.5 μm. Graphs show synapse number per 200 μm$^3$ at each distance (Two-Way ANOVA followed by Tukey's post-hoc test, n=6–7 animals per condition and 2–3 brain slices per animal).

The online version of this article includes the following figure supplement(s) for figure 6:

**Figure supplement 1.** In vivo DKK3 loss-of-function affects pre- and postsynaptic markers but does not affect Aβ plaque pathology or *Dkk1* expression in the mouse hippocampus.

suggesting that DKK3 is present in abnormally enlarged vesicles. This accumulation could indicate changes in DKK3 transport within dystrophic neurites affecting its degradation and/or secretion. We found that extracellular DKK3 levels are elevated in brain slices from J20 and NLGF models before substantial amyloid burden occurs, whereas acute exposure to Aβo increase both total and extracellular DKK3 levels in neurons. Intriguingly, a study reported reduced levels of DKK3 in the human AD brain and an AD mouse model (*Zhang et al., 2017*), but the specificity of the DKK3 antibody used in this study was not demonstrated. Moreover, this work indicated that overexpression of DKK3 restored memory in an AD model (*Zhang et al., 2017*). However, the generation of these mice was not fully characterized. Importantly, this study is in disagreement with other human proteomic studies (*Hesse et al., 2019*; *Johnson et al., 2022*; *Xu et al., 2019*) and our own findings that DKK3 is elevated in AD. Indeed, our studies using a validated antibody revealed that DKK3 is elevated in the brains of AD patients. In conclusion, our results are consistent with several other findings that DKK3 levels are increased in AD suggesting that elevated DKK3 may contribute to AD pathogenesis.

How are DKK3 levels regulated? Here we demonstrate that activity-dependent modulation of NMDARs regulates DKK3 levels. Several studies showed that Aβo block glutamate uptake by neurons, raising the extracellular glutamate levels and aberrantly activating NMDARs, leading to impaired synaptic function and memory (*Li et al., 2011*; *Mucke and Selkoe, 2012*). Furthermore, blockade of NMDARs protects synapse density and cognitive function in AD mouse models (*Hu et al., 2009*; *Ye et al., 2004*). Our studies revealed that the increased extracellular DKK3 levels in the hippocampus of J20 and NLGF mice are completely abolished by blockade of NMDARs. Conversely, extracellular DKK3 levels are increased by NMDAR-induced cLTD without affecting total levels of the protein. These apparently paradoxical results could be reconciled by our finding that only a small proportion of DKK3 is released from neurons, and therefore changes in DKK3 protein levels are not detected in the total cell lysate. These findings suggest that overactivation of NMDARs may trigger DKK3 release. Consistent with this suggestion, treatment with BFA, an inhibitor of vesicle transport used to study the release of proteins (*Brewer et al., 2022*; *He et al., 2015*; *Katsinelos et al., 2018*), blocks the increase in extracellular levels of DKK3 after cLTD. Given that LTD is increased in AD, these results suggest a possible mechanism for the regulation of DKK3 secretion in this condition.

DKK3 is the most highly expressed member of the DKK family of Wnt antagonists in the human and mouse brain (*Zhang et al., 2014*). Here, we demonstrate a novel role for DKK3 in differentially regulating both excitatory and inhibitory synapses in the hippocampus. These results are in contrast to those obtained with DKK1, which only affects excitatory synapses in the hippocampus (*Marzo et al., 2016*). Indeed, gain-of-function of DKK3 decreases the number of excitatory synapses but increases inhibitory synapses in the adult hippocampus. Conversely, in vivo knockdown of endogenous DKK3 in adult WT mice decreases inhibitory synapses but does not affect excitatory synapse density. This finding suggests that other molecules might compensate for the loss of DKK3 resulting in the maintenance of excitatory synapse number under basal conditions in the healthy brain. However, knockdown of DKK3 reduces the amplitude of mESPCs. A possible explanation for this finding is that endogenous DKK3 is required for excitatory synapse function without affecting their structural stability.

Our results also demonstrate that DKK3 signals through different pathways to regulate excitatory and inhibitory synapses. A key component of the canonical Wnt pathway is GSK3β. The activity of GSK3β is increased in the AD brain, which is associated with reduced Wnt signalling (*Leroy et al., 2007*). Importantly, activation of canonical Wnt pathway by inhibition of GSK3 blocks DKK3-mediated

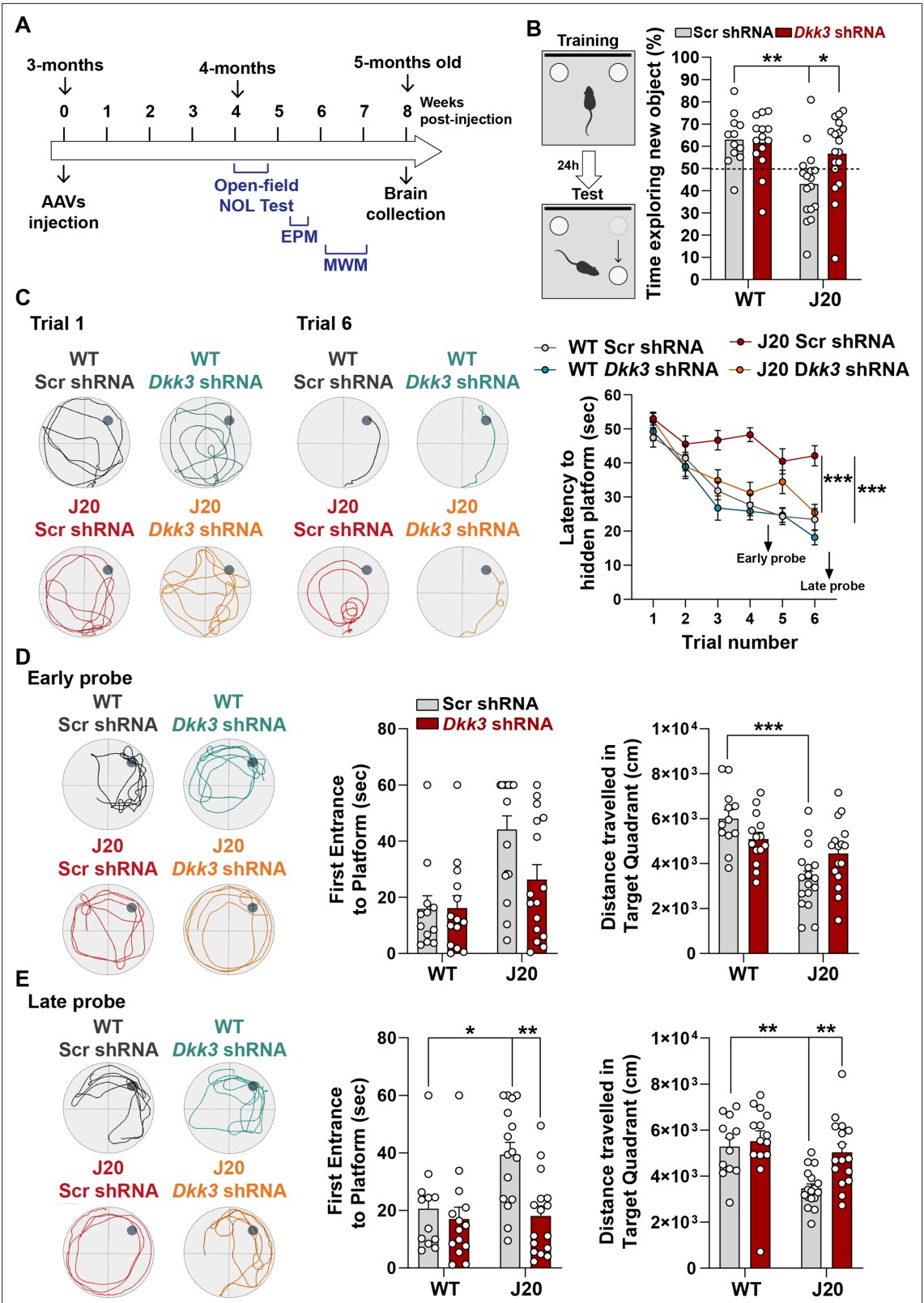

**Figure 7.** In vivo loss-of-function of DKK3 improves spatial memory in J20 mice. (**A**) Diagram depicting that 3-month-old WT and J20 mice were injected bilaterally with AAV9-Scr shRNA or AAV9-*Dkk3* shRNA in the CA3 area of the hippocampus. One month later, the behavior of animals was assessed using the Open-field, Novel Object Location (NOL) test, Elevated-Plus Maze (EPM), and the Morris water maze (MWM). (**B**) Novel Object Location Test. The percentage of time exploring the new object location *versus* the total time was evaluated (Two-Way ANOVA with Tukey's post-hoc test, n=12 WT

*Figure 7 continued on next page*

*Figure 7 continued*

Scr shRNA, 14 WT *Dkk3* shRNA, 17 J20 Scr shRNA, 16 J20 *Dkk3* shRNA). (**C– E**) Morris Water Maze. (**C**) Representative traces for the MWM Trials 1 and 6 are shown. Graph on the right shows the escape latency. Two-way ANOVA with repeated measures showed a significant effect over trials (animal group $F_{(3,55)}$ = 16.97, p-value <0.0001; trial $F_{(5,259)}$ = 42.94, p-value = 0.457; animal group and trial interaction $F_{(15,275)}$ = 2.753, p-value = 0.0006). For all analyses (n=12 WT Scr shRNA, 14 WT *Dkk3* shRNA, 17 J20 Scr shRNA, 16 J20 *Dkk3* shRNA). Graph show comparison between groups (Two-way ANOVA followed by Tukey's multiple comparisons). (**D, E**) Representative traces for the (**D**) Early and (**E**) Late probes. Graphs on the right show the time (s) to first reach the target location (Kruskal-Wallis followed by Dunns' multiple comparisons) and the distance (cm) traveled in the target quadrant (Two-way ANOVA followed by Tukey's post-hoc test for the early trial or Kruskal-Wallis followed by Dunns' multiple comparisons).

The online version of this article includes the following figure supplement(s) for figure 7:

**Figure supplement 1.** In vivo DKK3 loss-of-function does not affect locomotion or anxiety.

excitatory synapse loss, which is consistent with a role for DKK3 as an antagonist of canonical Wnt signaling (*Caricasole et al., 2003*; *Mizobuchi et al., 2008*; *Zhu et al., 2014*). GSK3β also plays a role in the production of mitochondrial ATP, a key event in maintaining synapses (*Gomez-Suaga et al., 2022*). However, this function is disrupted in the presence of toxic proteins such as Tau and, importantly, recovered by inhibition of GSK3β (*Gomez-Suaga et al., 2022*; *Szabo et al., 2023*). Thus, DKK3 could also contribute to the loss of excitatory synapses by impairing mitochondrial function through activation of GSK3β.

In contrast to excitatory synapses, blockade of GSK3 does not restore the impact of DKK3 on inhibitory synapse number. Instead, JNK blockade prevents DKK3-induced inhibitory synapse assembly, indicating a role for Wnt/JNK signaling in this process. Our finding that DKK3 activates JNK in the hippocampus is consistent with previous results in other cell types (*Abarzua et al., 2005*; *Mizobuchi et al., 2008*; *Yu et al., 2017*). Importantly, JNK blockade did not affect the loss of excitatory synapses by DKK3. Together, our results show that DKK3 regulates the stability of excitatory and inhibitory synapses in the adult hippocampus through different signaling pathways.

Our functional studies in the J20 mouse model of AD demonstrate that knocking down DKK3 in J20 mice ameliorates the changes in excitatory and inhibitory synapse number in the hippocampus both before and after plaque deposition. Although loss of synapses around amyloid plaques is well described (*Koffie et al., 2012*), changes in synapse number induced by silencing DKK3 is unlikely to be due to the formation of plaques as we observed similar synaptic changes when plaques are absent. In line with this view, knockdown of DKK3 does not affect the number or size of amyloid plaques in the J20 hippocampus. Moreover, our gain-of-function studies demonstrate that DKK3 directly affects the integrity of excitatory and inhibitory synapses. Crucially, downregulation of DKK3 also improves cognitive function, particularly spatial memory, in J20 mice. The rescue of the synaptic and cognitive defects is specific to DKK3 downregulation as no differences in *Dkk1* levels, a Wnt antagonist that affects synapses (*Marzo et al., 2016*), are observed.

Our functional studies in the J20 demonstrate a novel role for DKK3 in synaptic and cognitive function. In addition, our results using human AD brain samples provide strong support for the contribution of DKK3 to AD. Thus, DKK3 is a potential target for ameliorating excitatory and inhibitory synaptic impairment and memory dysfunction in AD.

## Materials and methods

### Human tissue

Anonymized human brain samples from control and AD patients were obtained from Cambridge Brain Bank (CBB), Division of the Human Research Tissue Bank, Addenbrooke's Hospital, Cambridge, UK. All samples were obtained with informed consent under CBB license (NRES 10/HO308/56) approved by the NHS Research Ethics Services. Tissues were stored at –80 °C. Demographic data and Braak stages for each subject are shown in *Supplementary file 1*.

### Mice

All procedures involving animals were conducted according to the Animals Scientific Procedures Act UK (1986) and in compliance with the ethical standards at University College London (UCL). WT C57BL/6 J were obtained from Jackson Laboratories. J20 mice were obtained from Jackson Laboratories and maintained on a C57BL/6 J genetic background. J20 hemizygous transgenic males were

bred with WT C57BL/6 J females to generate hemizygous transgenic mice (J20) and WT littermates. Genotyping was performed using DNA from ear biopsies and with the following primers to detect the human APP transgene: forward 5'- GGTGAGTTTGTAAGTGATGCC-3' and reverse 5'- TCTTCTTC TTCCACCTCAGC –3'. APP$^{NL-G-F/NL-G-F}$ mice were obtained from *Saito et al., 2014* and maintained in C57BL/6 J background as previously described (*Palomer et al., 2022*). Double transgenic mice (i*Dkk1*) were obtained by crossing tetO-Dkk1 transgenic mice with CaMKIIαrtTA2 transgenic mice (both C57BL/6 J background) as previously described (*Galli et al., 2014*; *Marzo et al., 2016*). Adult control (tetO-Dkk1, CaMKIIα-rtTA2 or wild-type littermates) and i*Dkk1* mice were fed with food pellets containing 6 mg/kg doxycycline for 14 days. Animals were housed in ventilated cages with access to food and water ad libitum and kept at 22 ± 2°C and 55 ± 10% humidity with a 12 hr/12 hr light cycle. The number of experimental animals, including males and females, was estimated based on previous experiments with expected similar effect size and variability. For viral injections and behavior analyses, mice of the same litter and similar age litters were randomly allocated into groups after genotyping. The ages of mice are specified in each figure legend, according to the experimental approach used.

## Primary hippocampal cultures treatments
Primary hippocampal neurons (700 cells/mm$^2$) were isolated from embryonic day 18 Sprague-Dawley rat embryos and cultured on poly-L-lysine coated plates or glass coverslips in Neurobasal medium containing N2 and B27 supplements (Invitrogen). Neurons were maintained in a 5% $CO_2$ humidified atmosphere at 37 °C. One-third of the media was replenished every seven days. All experiments were performed at 20–21 days-in-vitro (DIV).

## Aβ oligomers (Aβo) preparation
Synthetic Aβ (Aβ$_{1-42}$) or reverse Aβ (Aβ$_{42-1}$) peptides were prepared as previously described (*Purro et al., 2012*) with minor modifications. Briefly, HFIP films of Aβ$_{42-1}$ (Bachem, Cat# 4107743) and Aβ$_{1-42}$ (Bachem, Cat# 4090148) were dissolved in DMSO to a concentration of 5 mM. The solution was then sonicated at 40 Hz for 10 min followed by vortexing for 30 s. Sterile PBS was added to achieve a final Aβ$_{1-42}$ or Aβ$_{42-1}$ concentration of 100 µM, and vortexed again for 20 s. The peptides were left to oligomerize for 24 hr at 4 °C. Oligomeric preparations were centrifuged for 10 min at 14,000 × *g* and the solution was collected. Aggregation into oligomers was evaluated by native PAGE. Thirty µl of Aβ preparations were loaded into a 16% polyacrylamide gel and transferred onto a nitrocellulose membrane. Membranes were boiled for 5 min in TBS, blocked with 10% non-fat milk for 60 min at room temperature, and incubated with anti-Aβ antibody (6E10) O/N at 4 °C. 19-21DIV dissociated hippocampal neurons were treated with 200 nM Aβ$_{1-42}$ (monomers, dimers, trimers, and tetramers) or reverse Aβ$_{42-1}$ control for 3 hr at 37 °C in combination with APV (20 µM) or vehicle (PBS). Neurons were pre-treated with APV or vehicle 30 min prior to co-treatment with Aβ and APV.

## Chemical LTP and chemical LTD
Hippocampal neurons were subjected to cLTP or cLTD induction at 21DIV using glycine (*McLeod et al., 2018*) or NMDA (*Kamal et al., 1999*) respectively. Briefly, 200 µM glycine (Fisher Chemical) or 20 µM NMDA (Tocris Bioscience), or the vehicle PBS, were applied to cultures for 10 or 5 min, respectively. The media was then replaced with fresh medium. Lysates from neurons and extracellular media were processed for western blot analyses after 15 min of exposure to DMSO, NMDA, or glycine. Levels of phospho-GluA1 Ser845 and total GluA1 were evaluated as readouts.

## Hippocampal stereotactic surgery
Stereotactic injection of AAV9-EGFP-U6-Scramble shRNA or AAV9-EGFP-U6-*Dkk3* shRNA (both from VectorLabs) was performed bilaterally in the CA3 area of the hippocampus. The sequence for Scr shRNA and *Dkk3* shRNA were as follows: Scr shRNA, 5'-CCTAAGGTTAAGTCGCCCTCGCTCGAGC GAGGGCGACTTAACCTTAGGTTTTT-3' and *Dkk3* shRNA, 5'-GAGCCATGAATGTATCATTGACTC GAGTCAATGATACATTCATGGCTCTTTTT-3'. Adult mice were deeply anesthetized using a mixture of oxygen and isoflurane (4% for induction and 2–1% for maintaining anesthesia). Using a stereo-tactic frame, two injections were performed in the hippocampus at the following coordinates relative to bregma anteroposterior (AP) and mediolateral (ML) and to dural surface dorsoventral (DV): (1) –1.7AP,±2 ML, –1.85 DV; (2) –2.3 AP,±2.8 ML, –2.2 DV. Viral particles were injected into the brain using

a 10 µl Hamilton microliter syringe at an infusion rate of 100 nl/min. The needle was left for additional 5 min to ensure diffusion of the virus, then slowly retracted from the brain. After 4 weeks, synapses, behavior or gene expression were evaluated.

## Treatment of acute hippocampal slices

WT and J20 mouse brains were rapidly dissected and placed in 5% $CO_2$/95% $O_2$ ice-cold aCSF containing (in mM): 87 NaCl, 2.5 KCL, 25 $NaHCO_3$, 1.25 $Na_2HPO_4$, 0.5 $CaCl_2$, 7 $MgCl_2$, 10 D-(+)-Glucose, 75 sucrose (pH = 7.4). Sagittal 300 µm slices were obtained with a vibratome and transferred to 5% $CO_2$/95% $O_2$ aCSF (34 °C) containing (in mM): 125 NaCl, 2.5 KCL, 25 $NaHCO_3$, 1.25 $Na_2HPO_4$, 1 $CaCl_2$, 2 $MgCl_2$, 25 D-(+)-Glucose (pH = 7.4). Brain slices were maintained in warm aCSF solution for 60 min before starting treatments.

Brain slices were treated with 150 ng/ml recombinant DKK3 (R&D systems) or vehicle control (PBS) for 4 hr (for synapse density evaluation) or 60 min (for phospho-JNK levels) or 200 ng/ml for 3 hr (for electrophysiological recordings). Drugs used include 0.5 µM BIO (Calbiochem) or vehicle (DMSO); 60 nM CC-930 (Cayman Chemical) or vehicle (DMSO); 1 µM CHIR99021 (Calbiochem) or vehicle (DMSO); 20 µM NMDA (Tocris Bioscience) or vehicle (PBS); 50 µM APV (Tocris Bioscience) or vehicle (PBS); 10 µg/ml BFA (Biolegend) or vehicle (DMSO).

## Electrophysiology

Transverse hippocampal slices (300 µm) were cut on a vibratome in ice-cold aCSF bubbled with 95% $O_2$/5% $CO_2$ containing (in mM): 125 NaCl, 2.4 KCl, 26 $NaHCO_3$, 1.4 $NaH_2PO_4$, 20 D-(+)-Glucose, 0.5 $CaCl_2$ and 3 $MgCl_2$ as previously described (*Redlingshöfer et al., 2020*). CA3 pyramidal neurons were patched in whole-cell voltage-clamp configuration using pipettes (resistance 5–8 MΩ) pulled from borosilicate glass and filled with caesium gluconate intracellular solution containing (in mM): 130 D-gluconic acid lactone, 10 Hepes, 10 EGTA, 10 NaCl, 0.5 $CaCl_2$, 1 $MgCl_2$, 1 ATP and 0.5 GTP, 5 QX314 (pH to 7.2 with CsOH). Slices were perfused with the same aCSF solution as before except substituted with 1 mM $MgCl_2$ and 2 mM $CaCl_2$. All miniature currents were recorded in the presence of 100 nM TTX (Abcam). mEPSCs were held at –60 mV with 10 µM bicuculline (Tocris Bioscience) and 50 µM APV (Tocris Bioscience) added, whereas mIPSCs were held at 0 mV in the presence of 50 µM APV (*Ciani et al., 2011*). Currents were recorded using an Axopatch 200B amplifier and low pass filtered at 1 kHz and digitized (10 kHz). Analyses were performed using a combination of WinEDR and WinWCP (available free online at http://spider.science.strath.ac.uk/sipbs/software_ses.htm) software. For event detection, the "template" function in WinEDR software was used (*Clements and Bekkers, 1997*). Several filters were applied to exclude events that were unlikely to be genuine.

## Tissue processing for immunofluorescence microscopy

Acute slices (300 µm) were fixed in 4% paraformaldehyde (PFA)/4% sucrose for 20 min. Brains used for obtaining cryosection slices were fixed overnight in 4% PFA followed by cryopreservation in 30% sucrose before freezing. Free-floating sagittal hippocampal sections (30 µm) were obtained using a Leica cryostat.

Immunofluorescence staining of brain slices was performed as previously described (*Marzo et al., 2016*; *McLeod et al., 2018*). Briefly, slices were permeabilized and blocked using 10% donkey serum and 0.3% (for cryosections) or 0.5% (for acute slices) Triton X-100 in PBS for 3–5 hr at room temperature. Slices were then incubated with primary antibody overnight at 4 °C, followed by secondary antibody incubation (Alexa Fluor, 1:600, Invitrogen) for 2 hr at room temperature and DAPI (1:50,000, Invitrogen) staining for 10 min to counterstain nuclei. Thioflavin S (ThioS, Invitrogen) staining was performed as previously described (*Ly et al., 2011*). Briefly, after incubation with secondary antibodies (1:500-1:600, Alexa Fluor, Thermo Fisher Scientific or Jackson ImmunoResearch Labs), brain slices were dehydrated and incubated with 1% ThioS for 15 min. Slices were then rehydrated and washed in water. Brain slices were mounted with Fluoromount-G (Southern Biotech). TUNEL (Abcam) assay was performed following manufacturer's instructions.

Primary antibodies and dilutions used for immunofluorescence were: mouse Anti- Aβ (6E10 clone, 1:1000, Biolegend, Cat# 803001, RRID:AB_2564653), Rabbit Anti-Aβ (6E10 clone, chimeric, 1:1000, Novus Biologicals, Cat# NBP2-62566, RRID:AB_2917960), chicken anti-Bassoon (1:1,000, Synaptic Systems, Cat# 141 016, RRID:AB_2661779), Goat anti- Mouse DKK-3 (1:1000, R and D

Systems, Cat# AF948, RRID:AB_355734), Rabbit anti-Gephyrin (1:500, Synaptic Systems, Cat# 147 002, RRID:AB_2619838), Chicken anti-Glial Fibrillary Acidic Protein (GFAP) (1:500, Millipore, Cat# AB5541, RRID:AB_177521), Rabbit anti-Homer1 (1:1000, Synaptic Systems, Cat# 160 003, RRID:AB_887730), chicken Anti-GFP (1:500, Millipore, Cat# 06–896, RRID:AB_310288), rabbit anti-Iba1 (1:1000, FUJIFILM Wako Shibayagi, Cat# 019–1974,1 RRID:AB_839504), rat anti-LAMP1 (1:250, DSHB, 1D4B, RRID:AB_528127), rabbit Anti-NeuN (D3S3I) (1:1000, Cell Signaling Technology, Cat# 12943, RRID:AB_2630395), rabbit Anti-Neurofilament Heavy (1:5000, Abcam, Cat# ab8135, RRID:AB_306298), mouse Anti-PSD-95 (7E3-1B8) (1:500, Thermo Fisher Scientific, Cat# MA1-046, RRID:AB_2092361), guinea pig anti-vesicular GABA Transporter (vGAT) (1:500, Synaptic Systems, Cat# 131 004, RRID:AB_887873), guinea pig Anti-Vesicular Glutamate Transporter 1 (vGLUT1) (1:2000, Millipore, Cat# AB5905, RRID:AB_2301751), goat anti-Wnt7a/b (1:1000, R and D Systems, Cat# AF3460, RRID:AB_2304437), mouse Anti- beta Catenin (1:1000, BD Biosciences, Cat# 610153, RRID:AB_397554).

## Image acquisition and analyses

Confocal images were acquired using a Leica SP8 microscope and analyzed using ImageJ-FIJI (NIH) or Volocity 3D Image Analysis version 6.5.1 (Quorum Technologies). Analyses of synapses were performed blind to the genotype or group. For analyses of synaptic puncta, 3 images from at least 3 brain sections per animal or 8–12 images from 2 to 3 separate coverslips per culture were acquired using a 63 X (1.40 Numerical Aperture [NA]) oil objective. Each image comprised eight equidistant planes (0.3 µm apart) of 76nm x 76 nm. To analyze synapse density around a plaque, the plaque core was identified (core) and synapse number per 200 µm$^3$ was quantified at each distance from the plaque core. The number of pre- and post-synaptic puncta and number of synapses assessed as colocalization of pre- and post-synaptic markers was quantified using Volocity imaging software as previously described (*Galli et al., 2014*; *Marzo et al., 2016*; *McLeod et al., 2017*). Each independent value was obtained from the average of each brain section (unless otherwise stated) and the data were presented as relative values to the control and depicted as a percentage. For evaluating DKK3 in 6E10 plaques of J20 and NLGF mice, images from the whole hippocampus were obtained with a tile scan using a 20 X (0.75 NA) objective. Each stack comprised 30 equidistant planes 1 µm apart. Plaque quantification was performed blind to the group using ImageJ-FIJI as previously described (*Jones et al., 2023*). Images from the CA3 SR were thresholded, and the particle analysis tool was used to obtain the number and size of plaques and the percent coverage area of Aβ. For assessing DKK3 localization in ThioS positive plaques in the J20 brain, hippocampal images from at least 3 brain sections per animal were acquired using a 20 X (0.75 NA) objective. Each image comprised 21 equidistant planes 50 nm apart. Presence of DKK3 in plaques was evaluated manually, and the average of the different brain slices per mouse was obtained. Data were displayed as values relative to the control condition and depicted as a percentage. For evaluating the colocalization of DKK3 with different components of Aβ plaques, at least three plaques from three to four brain sections per animal were obtained using a 63 X (1.40 NA) oil objective. Each image comprised 8 equidistant planes (0.3 µm apart). Colocalization analyses of DKK3 and other markers (MAP2, Neurofilament H, IBA1, or GFAP) were performed in Volocity software using Pearson's coefficient tool based on intensity threshold values (the same threshold was used for all images). To assess DKK3 intensity in the CA3 area, three images from three brain slices per animal were taken using a 20 X (14 equidistant planes 2.41 µm apart) or a 40 X (8 equidistant planes 0.3 µm apart) oil objective. For each image, DKK3 intensity was normalized to MAP2 intensity using Volocity. The number of animals or independent cultures are indicated in the figure legends.

## Protein extraction and western blot

Proteins were extracted from WT and J20 hippocampi, primary hippocampal neurons, or human hippocampus with RIPA buffer (10 mM Tris, 100 mM NaCl, 1 mM EDTA, 1% Nonidet P-40, 0.1% SDS, 0.5% deoxycholate, pH = 8). Samples were then sonicated and centrifuged at 14,000 × *g* for 10 min at 4 °C. The supernatant representing the soluble protein fraction was collected. For human brain samples, the pellet was washed in RIPA buffer and solubilized in 4% SDS. Thereafter, samples were sonicated and centrifuged at 14,000 × *g* for 10 min at 4 °C to obtain the SDS-soluble fraction (insoluble protein fraction).

To evaluate the extracellular fraction of cultured neurons or brain slices, cell media from primary neurons and aCSF from acute slices treatments were collected and centrifuged at 14,000 × *g* for 10 min at 4 °C to evaluate extracellular levels of DKK3.

Protein concentration for all samples was quantified using a BCA kit (Thermo Fisher Scientific) according to the manufacturer's protocol. Protein extracts were resolved on 10% SDS-PAGE gels. Membranes (PVDF) were blocked with 5% non-fat milk and incubated with primary antibodies overnight at 4 °C, followed by incubation of secondary antibodies for 60 min at room temperature. Chemiluminescent images were acquired using ChemiDoc and fluorescent images (total JNK) were obtained using LiCor Odissey Clx. All images were quantified by densitometric analysis using ImageJ.

Primary antibodies and dilutions used for western blot were: Mouse Anti- Aβ (6E10 clone, 1:1000, Biolegend, Cat# 803001, RRID:AB_2564653), goat anti- Mouse DKK-3 (1:1000, R and D Systems, Cat# AF948, RRID:AB_355734), goat anti- Human DKK-3 (1:1000, R&D Systems, Cat# AF1118, RRID:AB_354610), rabbit anti-GAPDH (1:5000, Abcam, Cat# ab181602, RRID:AB_2630358), rabbit anti-GluA1 (1:1000, Cell Signaling Technology, Cat# 13185, RRID: AB_2732897), rabbit anti-phospho GluA1 Serine 845 (1:1000, Cell Signaling Technology, Cat# 8084, RRID: AB_10860773), rabbit anti-SAPK/JNK (1:1,000, Cell Signaling Technology, Cat# 9252, RRID:AB_2250373), mouse anti-Phospho-SAPK/JNK (Thr183/Tyr185) (1:500, Cell Signaling Technology, Cat# 9255, RRID:AB_2307321), mouse anti-Tubulin (1:5000, Sigma-Aldrich, Cat# T9026, RRID:AB_477593), mouse Anti-Vinculin (1:2000, Sigma Aldrich, Cat# v4505, RRID: AB_477617), HRP mouse anti-beta Actin (1:10,000, Abcam, Cat# ab8224, RRID:AB_449644).

Secondary antibodies and the dilutions used for western blot were: Donkey anti-goat IgG-HRP (1:2,000, Santa Cruz Biotechnology, Cat# sc-2020, RRID:AB_631728), donkey anti-goat IgG-HRP (1:10,000, R&D Systems, Cat# HAF109, RRID:AB_357236), Sheep anti-Mouse IgG-HRP (1:3000, GE Healthcare, Cat# NXA931, RRID:AB_772209), donkey anti-Rabbit IgG-HRP (1:2000, GE Healthcare, Cat# NA934, RRID:AB_772206), goat anti-Rabbit IgG IRDeye 800CW (1:10,000, Abcam, Cat# ab216773, RRID:AB_2925189).

## Western blot analyses

Chemiluminescent and fluorescent images from western blot membranes were acquired using ChemiDoc (Bio-Rad) or Odissey Clx (LiCor) respectively. DKK3 chemiluminescent signals from total homogenates or cell lysates were obtained within seconds (0.5–15 s), whereas extracellular signals were obtained within minutes (1–5 min). Blots were quantified by densitometric analyses using ImageJ. Densitometric target signals were normalized to the loading control signal (except for extracellular levels) and depicted as relative levels. The ratio of extracellular/lysate DKK3 was obtained by dividing the normalized values of DKK3 in the extracellular fraction by the values of the lysate for each condition. For quantification of P-JNK, densiometric quantification of P-JNK was corrected for the densitometric signal of total JNK.

## RNA extraction, reverse transcription and quantitative PCR (qPCR) analyses

RNA was extracted from the hippocampus of WT, J20, and i*Dkk*1 mice using TRIzol (Thermo Fisher Scientific) and the DirectZol RNA MiniPrep Kit (Zymo Research), following the manufacturer's instructions and as previously described (*Palomer et al., 2022*). Retrotranscription to first-strand cDNA was performed using the RevertAid H Minus First Strand cDNA Synthesis kit (Thermo Fisher Scientific) as per manufacturer's instructions. Five to 30 ng of the original RNA was used to perform qPCR for *Dkk3* and *Dkk1* using GoTaq qPCR Master Mix (Promega) in a CFX96 Bio-Rad system following the manufacturer's protocol (2 min at 95 °C followed by 40 cycles of denaturing at 95 °C and annealing/extension at 60 °C). *GusB*, *Pgk1,* and *Rpl13a* were used as housekeeping genes. All primers were purchased from Sigma-Aldrich and used at a final concentration of 0.5 µM. The following primers were used: *Dkk1 (forward:* 5'-CCGGGAACTACTGCAAAAAT-3'; reverse: 5'-AAAATGGCTGTGGTCAGAGG -3'), *Dkk3* (forward: 5'- GACCAGGGTGGGAAATAACA-3'; reverse: 5'-GACCACCTGTCCACTCTGGT -3'), *GusB* (forward: 5'-GGTTTCGAGCAGCAATGGTA-3'; reverse: 5'-GCTGCTTCTTGGGTGATGTC -3'), *Pgk1* (forward: 5'-TACCTGCTGGCTGGATGG-3'; reverse: 5'-CACAGCCTCGGCATATTTCT-3'), *Rpl13a* (forward: 5'-GACTCCTGGTGTGAACCCA-3'; reverse: 5'-CTCTACCCACAGGAGCAGT-3').

Relative expressions of *Dkk1* and *Dkk3* mRNAs were calculated using the comparative threshold cycle (Ct) method. Samples were run in triplicate and the average Ct values were obtained using the CFX Manager software version 3.1 (Bio-Rad). Gene expression was normalized to the expression of housekeeping genes.

### Behavioral tests

Two separate cohorts of 4-month-old WT and J20 mice injected with Scr or *Dkk3* shRNA were used to perform behavioral tests. All tests were carried out in a dimly lit room without noise interference. Animals were tracked using the automated SMART video tracking software (Panlab). Analyses were performed blind to the genotype and group.

### Elevated plus maze

Anxiety was tested using an elevated plus maze consisting of four arms (30.5x5 cm each arm), two of which were surrounded by walls (enclosed arms). The apparatus was elevated 40 cm above the ground. Each mouse was placed in the central square (neutral area) facing an open arm, and time spent in the open and enclosed arms was measured for 5 min.

### Open-field and novel object location test

Hippocampal-dependent spatial and recognition memory was tested using the Novel Object Location (NOL) test. The apparatus consisted of a square arena (45x45 cm), with cues in one of the walls. Mice were allowed to freely explore for 30 min to habituate them to the arena. Distance traveled and time spent in the center and the periphery were measured. On the second day, two identical objects were placed equidistantly from walls, and mice were allowed to explore for 10 min (NOL Acquisition). Twenty-four hours later, one of the objects was moved to a novel position, and mice were allowed to explore for 5 min (NOL Testing). Object preference was measured as the percentage of time exploring the novel object location.

### Morris water maze

Hippocampal-dependent spatial learning and memory were assessed using the MWM task as previously described (*Marzo et al., 2016*). In the first phase, mice performed four trials with a visible platform to check for deficiencies in vision or locomotion. The escape platform was made visible by using a high-contrast top surface and attaching a striped flag. In the second phase, mice were trained to find a hidden platform with extra-maze visible cues for 6 days with four trials per day. The platform was submerged 2 cm below water and placed at the midpoint of one of the quadrants. Each mouse was allowed to search for the platform for up to 60 s, after which mice that failed to reach the platform were placed on the platform. All mice were left on the platform for 10 s before they were returned to their home cage. Probe trials were conducted before the fifth day of training (early probe) and 24 hr after the last day of training (late probe). During the probe trials, the platform was removed from the pool, and mice were allowed to swim for 60 s.

### Human RNAseq analyses

The reprocessed ROSMAP (*De Jager et al., 2018*), MSBB (*Wang et al., 2018*) and MayoRNAseq (*Allen et al., 2016*) temporal cortex RNASeq datasets and their associated phenotypic data such as Braak (*Braak et al., 2006*) and CERAD (*Mirra et al., 1991*) scores were downloaded from the AMP-AD consortium. For the ROSMAP study, AD cases were defined as individuals with a cognitive diagnosis of AD with no other cause of cognitive impairment (cogdx = 4 and cogdx = 5), and controls were defined as those with no cognitive impairment (cogdx = 1). For the MSBB dataset, controls were defined as those with a CERAD score of 1 (normal) and a clinical dementia rating (CDR) of 0 or 0.5 (no cognitive deficits or questionable dementia respectively), whereas cases were defined as subjects with CERAD score of 2, 3, or 4 (possible, probable, definite AD) and a CDR of 2 or greater (mild dementia, moderate dementia, or severe to terminal dementia). For the MayoRNASeq dataset, individuals were already classified as an AD case or control based on neuropathology. All cases had a Braak stage of IV or greater. Controls had a Braak stage of III or lower.

RNASeq datasets from ROSMAP (*De Jager et al., 2018*), MSBB (*Wang et al., 2018*), and MayoR-NAseq (*Allen et al., 2016*) underwent quality control using RNASeQC (*DeLuca et al., 2012*) and

were normalized for gene length and GC content with low expressed genes filtered out. Following quality control and normalization, 16,485 genes remained in the analysis. Linear mixed effect models (LMEM) in combination with principal component (PC) analyses were performed on normalized counts to combine data and adjust for batch effects and hidden confounders. LMEM used sex, age at death, and the first three PCs as fixed effects, whilst individual ID and sequencing batch were used as random effects. Logistic regression was then performed on residuals from the LMEM for AD case/control status (n=379 AD cases, 248 controls). Ordinal regressions were also performed on residuals from the LMEM for Braak stage (0–6) (n=627) and CERAD scores (1-4) (n=537). The β-coefficient indicates the degree of differential *DKK3* expression.

## Statistical analyses

All graphed data are displayed as mean ± SEM. Statistical analyses were performed using GraphPad Prism version 8.0.2. Statistical outliers were determined using Grubbs and ROUT tests. Dataset normality was tested by the D'Agostino and Pearson or Shapiro-Wilk tests. When datasets passed normality, comparisons between two groups were analyzed using the unpaired two-sided Student's T-test whilst comparisons between more than two groups used one- or two-way ANOVA, followed by Tukey's multiple comparisons tests. For non-normally distributed data, comparisons between two groups were performed using the Mann-Whitney U test, and comparisons between more than two groups with Kruskal-Wallis followed by Dunn's multiple comparison test. Pearson correlation coefficient was used for colocalization analyses. In all graphs, N-numbers corresponding to the number of independent primary cultures, animals, or human subjects, unless otherwise specified, are shown. In all figures, p-values are depicted as: *p-value ≤0.05, **p-value ≤0.01, ***p-value ≤0.001.

## Acknowledgements

We thank Professors Takashi Saito and Takaomi Saido for the APP[NL-G-F] mice and Professors Qingbo Xu for providing us with brain samples of the *DKK3*[-/-] *ApoE*[-/-] mice used as controls for antibody validation. We thank members of the Salinas lab and our collaborators Professors Francesca Cacucci and Alasdair Gibb for their support and discussion on this project. We also thank Dr Ernest Palomer for his technical advice on qPCR analyses. The results published here are in whole or in part based on data obtained from the AD Knowledge Portal (https://adknowledgeportal.org). ROSMAP data were provided by the Rush Alzheimer's Disease Center, Rush University Medical Center, Chicago. Data collection was supported through funding by NIA grants P30AG10161 (ROS), R01AG15819 (ROSMAP; genomics and RNAseq), R01AG17917 (MAP), R01AG30146, R01AG36042 (5hC methylation, ATACseq), RC2AG036547 (H3K9Ac), R01AG36836 (RNAseq), R01AG48015 (monocyte RNAseq) RF1AG57473 (single nucleus RNAseq), U01AG32984 (genomic and whole exome sequencing), U01AG46152 (ROSMAP AMP-AD, targeted proteomics), U01AG46161(TMT proteomics), U01AG61356 (whole genome sequencing, targeted proteomics, ROSMAP AMP-AD), the Illinois Department of Public Health (ROSMAP), and the Translational Genomics Research Institute (genomic). Additional phenotypic data can be requested at https://www.radc.rush.edu/. MSBB data were generated from postmortem brain tissue collected through the Mount Sinai VA Medical Center Brain Bank and were provided by Dr. Eric Schadt from Mount Sinai School of Medicine. The Mayo RNAseq study data was led by Dr. Nilüfer Ertekin-Taner, Mayo Clinic, Jacksonville, FL as part of the multi-PI U01 AG046139 (MPIs Golde, Ertekin-Taner, Younkin, Price). Samples were provided from the following sources: The Mayo Clinic Brain Bank. Data collection was supported through funding by NIA grants P50 AG016574, R01 AG032990, U01 AG046139, R01 AG018023, U01 AG006576, U01 AG006786, R01 AG025711, R01 AG017216, R01 AG003949, NINDS grant R01 NS080820, CurePSP Foundation, and support from Mayo Foundation. Study data includes samples collected through the Sun Health Research Institute Brain and Body Donation Program of Sun City, Arizona. The Brain and Body Donation Program is supported by the National Institute of Neurological Disorders and Stroke (U24 NS072026 National Brain and Tissue Resource for Parkinsons Disease and Related Disorders), the National Institute on Aging (P30 AG19610 Arizona Alzheimers Disease Core Center), the Arizona Department of Health Services (contract 211002, Arizona Alzheimers Research Center), the Arizona Biomedical Research Commission (contracts 4001, 0011, 05–901 and 1001 to the Arizona Parkinson's Disease Consortium) and the Michael J Fox Foundation for Parkinsons Research. Diagrams included

in the figures were created with Biorender.com. This work was funded by Alzheimer's Society (AS-PG-17–006), MRC (MR/S012125/1, MR/M024083/1, and MR/X010589/1), Alzheimer's Research UK (ARUK-PG2018A-002).

## Additional information

### Funding

| Funder | Grant reference number | Author |
| --- | --- | --- |
| Alzheimer's Society | AS-PG-17-006 | Patricia C Salinas |
| Medical Research Council | MR/S012125/1 | Patricia C Salinas |
| Medical Research Council | MR/M024083/1 | Patricia C Salinas |
| Alzheimer's Research UK | ARUK-PG2018A-002 | Patricia C Salinas |
| Medical Research Council | MR/X010589/1 | Patricia C Salinas |

The funders had no role in study design, data collection and interpretation, or the decision to submit the work for publication.

### Author contributions

Nuria Martin Flores, Marina Podpolny, Data curation, Software, Formal analysis, Investigation, Visualization, Methodology, Writing – original draft, Writing – review and editing; Faye McLeod, Data curation, Software, Formal analysis, Investigation, Visualization, Methodology, Writing – review and editing; Isaac Workman, Investigation, Visualization, Writing – review and editing; Karen Crawford, Dobril Ivanov, Ganna Leonenko, Software, Formal analysis; Valentina Escott-Price, Software, Investigation, Writing – review and editing; Patricia C Salinas, Conceptualization, Resources, Supervision, Funding acquisition, Visualization, Methodology, Writing – original draft, Project administration, Writing – review and editing

### Author ORCIDs

Nuria Martin Flores  https://orcid.org/0000-0003-2537-751X
Patricia C Salinas  http://orcid.org/0000-0002-5748-083X

### Ethics

Human subjects: Anonymized human brain samples from control and AD patients were obtained from Cambridge Brain Bank (CBB), Division of the Human Research Tissue Bank, Addenbrooke's Hospital, Cambridge, UK. All samples were obtained with informed consent under CBB license (NRES 10/HO308/56) approved by the NHS Research Ethics Services.

All procedures involving animals were conducted according to the Animals Scientific Procedures Act UK (1986) and in compliance with the ethical standards at University College London (UCL).

Reviewer #1 (Public Review): https://doi.org/10.7554/eLife.89453.3.sa1
Reviewer #2 (Public Review): https://doi.org/10.7554/eLife.89453.3.sa2
Author Response https://doi.org/10.7554/eLife.89453.3.sa3

## Additional files

### Supplementary files

• Supplementary file 1. Human brain samples information. M=Male, F=Female, PMI = Post-mortem interval.

• MDAR checklist

### Data availability

All data generated or analysed during this study are included in the manuscript and supporting files.

The following previously published datasets were used:

| Author(s) | Year | Dataset title | Dataset URL | Database and Identifier |
|---|---|---|---|---|
| De Jager PL, Ma Y, McCabe C, Xu J, Vardarajan BN, Felsky D, Klein H-U, White CC, Peters MA, Lodgson B, Nejad P, Tang A, Mangravite LM, Yu L, Gaiteri C, Mostafavi S, Schneider JA, Bennett DA | 2018 | The RNAseq Harmonization Study (rnaSeqReprocessing), ROSMAP | https://doi.org/10.7303/syn9702085 | Synapse, 10.7303/syn9702085 |
| Wang M, Beckmann ND, Roussos P, Wang E, Zhou X, Wang Q, Ming C, Neff R, Ma W, Fullard JF, Hauberg ME, Bendl J, Peters MA, Logsdon B, Wang P, Mahajan M, Mangravite LM, Dammer EB, Duong DM, Lah JJ, Seyfried NT, Levey AI, Buxbaum JD, Ehrlich M, Gandy S, Katsel P, Haroutunian V, Schadt E, Zhang B | 2018 | The RNAseq Harmonization Study (rnaSeqReprocessing), MSBB | https://doi.org/10.7303/syn9702085 | Synapse, 10.7303/syn9702085 |
| Allen M, Carrasquillo MM, Funk C, Heavner BD, Zou F, Younkin CS, Burgess JD, Chai H-S, Crook J, Eddy JA, Li H, Logsdon B, Peters MA, Dang KK, Wang X, Serie D, Wang C, Nguyen T, Lincoln S, Malphrus K, Bisceglio G, Li M, Golde TE, Mangravite LM, Asmann Y, Price ND, Petersen RC, Graff-Radford NR, Dickson DW, Younkin SG, Ertekin-Taner N | 2016 | The RNAseq Harmonization Study (rnaSeqReprocessing), MayoRNAseq | https://doi.org/10.7303/syn9702085 | Synapse, 10.7303/syn9702085 |

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

# Appendix 1

## Appendix 1—key resources table

| Reagent type (species) or resource | Designation | Source or reference | Identifiers | Additional information |
|---|---|---|---|---|
| Strain, strain background (Mouse) | C57BL/6 J | The Jackson Laboratory | JAX: 000664 RRID: MGI:2159769 | WT mice |
| Strain, strain background (Mouse) | B6.Cg-Zbtb20$^{Tg(PDGFB-APPSwInd)20Lms}$/2Mmjax (hAPP-J20) | The Jackson Laboratory | | J20 mice |
| Strain, strain background (Mouse) | App$^{tm3.1Tcs}$/App$^{tm3.1Tcs}$ (APP$^{NL-G-F}$) | Professor Takaomi Saido | *Saito et al., 2014* | NLGF mice |
| Biological sample (Rat) | Primary neurons | Charles River | RRID: MGI:5651135 | Freshly isolated from Sprague-Dawley E18 embryos |
| Biological sample (Mouse) | *Dkk3* knock-out brain tissue | Professor Qingbo Xu | *Yu et al., 2017* | |
| Biological sample (Human) | Control and patient brain tissue | Cambridge Brain Bank (CBB), Division of the Human Research Tissue Bank, Addenbrooke's Hospital, Cambridge, UK | | |
| Antibody | Mouse monoclonal Anti-Aβ (6E10 clone) | Biolegend | Cat# 803001, RRID: AB_2564653 | 1:1,000 for IF and WB |
| Antibody | Rabbit monoclonal Anti-Aβ (6E10 clone, chimeric) | Novus Biologicals | Cat# NBP2-62566, RRID: AB_2917960 | 1:1,000 for IF |
| Antibody | Chicken polyclonal anti-Bassoon | Synaptic Systems | Cat# 141 016, RRID: AB_2661779 | 1:1,000 for IF |
| Antibody | Goat polyclonal anti-Mouse DKK-3 Systems | R&D Systems | Cat# AF948, RRID: AB_355734 | 1:1,000 for IF and WB |
| Antibody | Goat polyclonal anti-Human DKK-3 | R&D Systems | Cat# AF1118, RRID: AB_354610 | 1:1,000 for WB |
| Antibody | Rabbit monoclonal anti-GAPDH | Abcam | Cat# ab181602, RRID: AB_2630358 | 1:5,000 for WB |
| Antibody | Rabbit polyclonal anti-Gephyrin | Synaptic Systems | Cat# 147 002, RRID: AB_2619838 | 1:500 for IF |
| Antibody | Chicken polyclonal anti-Glial Fibrillary Acidic Protein (GFAP) | Millipore | Cat# AB5541, RRID: AB_177521 | 1:500 for IF |
| Antibody | Rabbit monoclonal anti-GluA1 | Cell Signaling Technology | Cat# 13185, RRID: AB_2732897 | 1:1,000 for WB |
| Antibody | Rabbit monoclonal anti-phospho GluA1 Serine 845 | Cell Signaling Technology | Cat# 8084, RRID: AB_10860773 | 1:1,000 for WB |
| Antibody | Chicken polyclonal Anti-GFP | Millipore | Cat# 06–896, RRID: AB_310288 | 1:500 for IF |
| Antibody | Rabbit polyclonal anti-Homer1 | Synaptic Systems | Cat# 160 003, RRID: AB_887730 | 1:1,000 for IF |
| Antibody | Rabbit polyclonal anti-Iba1 | FUJIFILM Wako Shibayagi | Cat# 019–1974,1 RRID: AB_839504 | 1:1,000 for IF |
| Antibody | Rabbit polyclonal anti-SAPK/JNK | Cell Signaling Technology | Cat# 9252, RRID: AB_2250373 | 1:1,000 for WB |
| Antibody | Mouse monoclonal anti-Phospho-SAPK/JNK (Thr183/Tyr185) | Cell Signaling Technology | Cat# 9255, RRID: AB_2307321 | 1:500 for WB |
| Antibody | Rat monoclonal anti-LAMP1 | DSHB | Cat# 1D4B, RRID: AB_2134500 | 1:250 for IF |
| Antibody | Rabbit monoclonal Anti-NeuN (D3S3I) | Cell Signaling Technology | Cat# 12943, RRID: AB_2630395 | 1:1,000 for IF |
| Antibody | Rabbit polyclonal Anti-Neurofilament Heavy | Abcam | Cat# ab8135, RRID: AB_306298 | 1:5,000 for IF |

*Appendix 1 Continued on next page*

*Appendix 1 Continued*

| Reagent type (species) or resource | Designation | Source or reference | Identifiers | Additional information |
|---|---|---|---|---|
| Antibody | Mouse monoclonal Anti-PSD-95 (7E3-1B8) | Thermo Fisher Scientific | Cat# MA1-046, RRID: AB_2092361 | 1:500 for IF |
| Antibody | Mouse monoclonal anti-Tubulin | Sigma-Aldrich | Cat# T9026, RRID: AB_477593 | 1:5,000 for WB |
| Antibody | Guinea pig polyclonal anti-vesicular GABA Transporter (vGAT) | Synaptic Systems | Cat# 131 004, RRID: AB_887873 | 1:500 for IF |
| Antibody | Guinea pig polyclonal Anti-Vesicular Glutamate Transporter 1 (vGLUT1) | Millipore | Cat# AB5905, RRID: AB_2301751 | 1:2,000 for IF |
| Antibody | Mouse monoclonal Anti-Vinculin | Sigma Aldrich | Cat# v4505, RRID: AB_477617 | 1:2,000 for WB |
| Antibody | Goat polyclonal anti-Wnt7a/b | R&D Systems | Cat# AF3460, RRID: AB_2304437 | 1:1,000 for IF |
| Antibody | Mouse monoclonal Anti-beta Catenin | BD Biosciences | Cat# 610153, RRID: AB_397554 | 1:1,000 for IF |
| Antibody | HRP mouse monoclonal anti-beta Actin | Abcam | Cat# ab8224, RRID: AB_449644 | 1:10,000 for WB |
| Sequence-based reagent | shRNA Scrambled sequence | This paper | shRNA Scrambled sequence | 5'-CCTAAGGTTAAGTCGCCC TCGCTCGAGCGAGGGCGAC TTAACCTTAGGTTTTT-3' |
| Sequence-based reagent | shRNA *Dkk3* sequence | This paper | shRNA *Dkk3* sequence | 5'-GAGCCATGAATGTATCATT GACTCGAGTCAATGATACATT CATGGCTCTTTTT-3 |
| Peptide, recombinant protein | Recombinant DKK3 protein | R&D Systems | Cat# 1118-DK | 150 ng/ml or 200 ng/ml |
| Peptide, recombinant protein | HFIP-treated Amyloid β 1–42 peptide | Bachem | Cat# 4090148 | 200 nM |
| Peptide, recombinant protein | HFIP-treated Reverse Amyloid β 42–1 | Bachem | Cat# 4107743 | 200 nM |
| Commercial assay or kit | TUNEL assay | Abcam | Cat# ab66110 | |
| Commercial assay or kit | BCA kit | Thermo Fisher Scientific | Cat# 23225 | |
| Commercial assay or kit | DirectZol Miniprep RNA Kit | Zymo Research | Cat# R2052 | |
| Commercial assay or kit | RevertAid H Minus First Strand cDNA Synthesis kit | Thermo Fisher Scientific | Cat# K1632 | |
| Commercial assay or kit | GoTaq qPCR Master Mix | Promega | Cat# A6002 | |
| Chemical compound, drug | Fluoromount-G | Southern Biotech | Cat# 0100–01 | |
| Chemical compound, drug | FluorSave Reagent | Merck Millipore | Cat# 345789 | |
| Chemical compound, drug | TRIzol Reagent | Thermo Fisher Scientific | Cat# 15596018 | |
| Chemical compound, drug | DAPI | Thermo Fisher Scientific | Cat# 62248 | 1:50,000 dilution |
| Chemical compound, drug | Thioflavin S | Sigma Aldrich | Cat# T1892 | 1% dilution |
| Chemical compound, drug | BIO | Calbiochem | Cat# 361550 | 0.5 µM |
| Chemical compound, drug | CC-930 | Cayman Chemical | Cat# CAY22466 | 60 nM |

*Appendix 1 Continued on next page*

*Appendix 1 Continued*

| Reagent type (species) or resource | Designation | Source or reference | Identifiers | Additional information |
|---|---|---|---|---|
| Chemical compound, drug | CHIR99021 | Calbiochem | Cat# 361559 | 1 µM |
| Chemical compound, drug | NMDA | Tocris Bioscience | Cat# 0114 | 20 µM |
| Chemical compound, drug | Glycine | Fisher Chemical | Cat# 10070150 | 200 µM |
| Chemical compound, drug | D-APV | Tocris Bioscience | Cat# 0106 | 20 µM or 50 µM |
| Chemical compound, drug | Brefeldin A | Biolegend | Cat# 420601 | 10 µg/ml |
| Software, algorithm | ImageJ-FIJI | NIH | https://imagej.net/software/fiji/ RRID:SCR_001935 | |
| Software, algorithm | GraphPad Prism version 8 | GraphPad software | https://www.graphpad.com/scientific-software/prism RRID:SCR_002798 | |
| Software, algorithm | Image Lab | Bio-Rad | https://www.bio-rad.com/en-uk/product/image-lab-software?ID=KRE6P5E8Z RRID:SCR_014210 | |
| Software, algorithm | Volocity 3D Image Analysis version 6.5.1 | Quorum Technologies | https://www.volocity4d.com/ RRID:SCR_002668 | |
| Software, algorithm | CFX Manager Software 3.1 | Bio-Rad | https://www.bio-rad.com/en-uk/sku/1845000-cfx-manager-software?ID=1845000 | |
| Software, algorithm | SMART Video tracking | Panlab | https://www.panlab.com/en/products/smart-video-tracking-software-panlab RRID:SCR_002852 | |
| Software, algorithm | WinEDR | University of Strathclyde | http://spider.science.strath.ac.uk/sipbs/software_ses.htm | |
| Software, algorithm | WinWCP | University of Strathclyde | http://spider.science.strath.ac.uk/sipbs/software_ses.htm RRID:SCR_014713 | |
| Software, algorithm | RNASeQC | *DeLuca et al., 2012* | https://software.broadinstitute.org/cancer/cga/rna-seqc | |
| Other | Confocal microscope | Leica | SP8 | Materials and Methods, Image acquisition and analyses |
| Other | Vibratome | Leica | VT100S | Materials and Methods, Treatment of acute hippocampal slices and Electrophysiology |
| Other | Cryostat | Leica | CM1850 | Materials and Methods, Tissue processing for immunofluorescence microscopy |
| Other | Patch-clamp amplifier | Molecular Devices | Axopatch 200B | Materials and Methods, Electrophysiology |

