## [Editor Report · eLife assessment]

This **important** manuscript investigates the roles of DKK3 in AD synapse integrity. Although previous work has identified the involvement of Wnt and DKK1 in synaptic physiology, this study provides **compelling** evidence that suppression of DKK3 rescues the changes in excitatory synapse numbers, as well as memory deficits in an established AD model mice. The authors provide both gain and loss of function data that support the main conclusion and advance our understanding of the mechanisms by which Wnt pathway mediates early synaptic dysfunction in AD models.

---

## [Referee Report · Reviewer #1 (Public Review)]

In this study, Nuria Martin-Flores, Marina Podpolny and colleagues investigate the role of Dickkopf-3 (DKK3), a Wnt antagonist in synaptic dysfunction in Alzheimer's disease. Loss of synapses is a feature of Alzheimer's and other forms of dementia such as frontotemporal dementia and linked amyotrophic lateral sclerosis (FTD). The authors utilise a broad range of experimental approaches. They show that DKK3 levels are increased in Alzheimer's disease and that this occurs early in disease. This is an important finding since early disease changes are believed to be the most important. They also show increases in DKK3 in transgenic mouse models of Alzheimer's disease and that DKK3 knockdown restores synapse number and memory in one such model. Finally, they link these DKK3 increases to loss of excitatory synapses via the blockade of the Wnt pathway and subsequent activation of GSK3B; GSK3B is strongly linked to both Alzheimer's disease and FTD. The quality of the data is good and the conclusions well supported by these data. There are no major weaknesses. The findings support studies that target the Wnt pathway as a potential therapeutic for Alzheimer's disease.

---

## [Referee Report · Reviewer #2 (Public Review)]

Summary:

This manuscript by Martin-Flores et al. examines the role of DKK3 in Alzheimer's disease, focusing on the regulation of synaptic number. Using human AD brain databases and tissue samples, the authors demonstrate increased levels of DKK3 protein and mRNA in the brains of AD patients. DKK3 is expressed in excitatory neurons in WT mouse brains and accumulates at atrophic neurites around amyloid plaques in AD mouse brains. Interestingly, the secretion of DKK3 appears to be regulated by NMDAR antagonists, as well as chemical LTD. Through gain and loss of function studies, the authors reveal that DKK3 regulates the number of both excitatory and inhibitory synapses with distinct downstream pathways. Finally, the authors investigate the contribution of DKK3 to synaptic changes in AD and find that DKK3 loss of function rescues both excitatory and inhibitory synaptic defects, resulting in improved memory function in J20 mice.

Strengths:

Overall, the data is clearly presented and deals with the novel roles of DKK3 in controlling excitatory and inhibitory synapses. The finding that shRNA expression of DKK3 in AD model mice rescues synaptic phenotypes and memory impairment is potentially interesting and may provide a new strategy for AD treatment.

Weaknesses:

There are no major weaknesses.

---

## [Author Response]

The following is the authors’ response to the original reviews.

**eLife Assessment**
This valuable manuscript investigates the roles of DKK3 in AD synapse integrity. Although previous work has identified the involvement of Wnt and DKK1 in synaptic physiology, this study provides compelling evidence that suppression of DKK3 rescues the changes in excitatory synapse numbers, as well as memory deficits in an established AD model mice. The authors provide both gain and loss of function data that support the main conclusion and advance our understanding of the mechanisms by which Wnt pathway mediates early synaptic dysfunction in AD models.
**Public Reviews:**

**Reviewer #1 (Public Review):**

In this study, Nuria Martin-Flores, Marina Podpolny and colleagues investigate the role of Dickkopf-3 (DKK3), a Wnt antagonist in synaptic dysfunction in Alzheimer's disease. Loss of synapses is a feature of Alzheimer's and other forms of dementia such as frontotemporal dementia and linked amyotrophic lateral sclerosis (FTD). The authors utilise a broad range of experimental approaches. They show that DKK3 levels are increased in Alzheimer's disease and that this occurs early in disease. This is an important finding since early disease changes are believed to be the most important. They also show increases in DKK3 in transgenic mouse models of Alzheimer's disease and that DKK3 knockdown restores synapse number and memory in one such model. Finally, they link these DKK3 increases to loss of excitatory synapses via the blockade of the Wnt pathway and subsequent activation of GSK3B; GSK3B is strongly linked to both Alzheimer's disease and FTD. The quality of the data is good and the conclusions well supported by these data. There are no major weaknesses. The findings support studies that target the Wnt pathway as a potential therapeutic for Alzheimer's disease.

**Reviewer #2 (Public Review):**
This manuscript by Martin-Flores et al., has examined the role of DKK3 in Alzheimer's disease, focusing on the regulation of synaptic numbers. By using human AD brain databases and tissue samples, the authors showed that DKK3 protein and mRNA levels are increased in the brains of AD patients. DKK3 is expressed in the excitatory neurons in WT mouse brains and accumulates at atrophic neurites around amyloid plaques in AD mouse brains. Interestingly, secretion of DKK3 appears to be regulated by NMDAR antagonist as well as chemical LTD. Through gain and loss of function studies, the authors showed that DKK3 regulates the number of excitatory as well as inhibitory synapses with distinct downstream pathways. Finally, the authors investigated the contribution of DKK3 to synaptic changes in AD and found that DKK3 loss of function rescues both the excitatory and inhibitory synaptic defects, resulting in the improvement of memory function in J20 mice.Overall, the data is clearly presented and deals with novel roles of DKK3 in controlling excitatory and inhibitory synapses. The finding that shRNA expression of DKK3 in AD model mice rescues synaptic phenotypes and memory impairment is potentially interesting and may provide a new strategy for AD treatment.

We would like to thank the Editors and the Reviewers for their very insightful suggestions. We are delighted to receive very positive reviews of our manuscript. In response to the comments made by the reviewers, we have carried out an extensive revision of our manuscript. In the revised manuscript, we have addressed all the comments made by the reviewers.

**Recommendations for the authors:**

**Reviewer #1:**
My only comment regards the role of GSK3B activation in synaptic dysfunction and its targets. GSK3B is a Tau kinase but is also involved in IP3 receptor delivery of Ca2+ to mitochondria. This delivery is major regulator of mitochondrial ATP production and synaptic function is heavily dependent on ATP. Both Alzheimer's disease and FTD insults have been linked to GSK3B activation -see for e.g. Szabo EMBO R 2023, Gomez-Suaga Aging Cell 2022. It might be valuable to readers for the authors to speculate briefly on potential GSK3B synaptic targets in the Discussion.

We appreciate the reviewer for this suggestion. In the Discussion, we now included how GSK3β may contribute to synaptic dysfunction and loss in the context of increased DKK3 levels and in Alzheimer’s disease.

**Reviewer #2:**
1. In Fig 1B, the authors showed that soluble DKK3 levels were increased in Braak 1-3 patients, while no changes were observed in Braak 4-5. If the secretion of DKK3 is dependent on NMDAR activity, does this data imply that Braak 4-5 patients have reduced NMDAR activity in general, resulting in the reduced DKK3 release even with the increased mRNA levels? It would be interesting to test this hypothesis in a mouse AD model.

In Figure 1B, we analyzed the levels of soluble and insoluble DKK3 in the hippocampus of AD patients at different disease stages based on their Braak stages. As the reviewer indicated, soluble levels of DKK3 were increased in patients with Braak I-III but not at later stages. Importantly, DKK3 levels were also elevated in Braak IV-VI patients, but only in the insoluble fraction (Figure 1C), suggesting that DKK3 could accumulate within Aβ aggregates. Based on these findings, we cannot conclude that DKK3 release is reduced at later stages of the disease in patients.

To explore the underlying mechanisms regulating DKK3 levels, we used cultured hippocampal neurons and AD mouse brain slices. In mouse models, we have demonstrated that extracellular DKK3 levels (secreted DKK3 fraction) depends on NMDAR activation early in the disease progression (Figure 2E, F). Moreover, we also provide new data showing that antagonizing NMDAR partially blocks the increase of DKK3 extracellular levels induced by oligomeric Aβ (see response to question 4 of this reviewer and Figure S2G, H, current Figure 2 figure supplement 2 G, H). It is well established that oligomeric Aβ promotes hyperexcitability through, in part, the aberrant activation of NMDAR (Li S et al., 2011, PMID: 21543591; Mucke L and Selkoe DJ et al., 2012, PMID: 22762015). In line with this, NMDAR blockers prevent Aβ-induced synapse loss and improve cognition in AD models (Hu NW et al., 2009, PMID: 19918059; Ye C et al., 2004, PMID: 15288443). In addition, an NMDAR antagonist is currently approved as a drug treatment for AD patients (Cumming J 2021, PMID: 33441154). Together, our findings in dissociated neurons, AD mouse brain and human samples indicate that soluble Aβ oligomers promote the release of DKK3 through NMDAR activation and suggest that this mechanism might also be occurring in the brain of AD patients.

2. Recent work (Yuan et al., 2022, Nature) has shown that dystrophic neurites/axonal spheroids found around Aβ deposits are filled with neuronal endolysosomes. Are DKK3 in ThioS positive amyloid plaques located in endolysosomes of these axonal spheroids? If so, does this data mean that DKK3 in Fig 2B-D represents the entrapped DKK3 protein population that fails to be secreted from dystrophic neurites?

The reviewer points an interesting question. Our results show that secretion of DKK3 is increased in two AD models before substantial plaque load. Later in the disease, DKK3 accumulates in dystrophic neurites (visualized as axonal spheroids) surrounding amyloid plaques. To address if DKK3 protein is located in vesicles of the endolysosomal pathway within axonal spheroids, we performed co-localization analyses of DKK3 and the endolysosomal marker LAMP1. We found that DKK3 colocalized with LAMP1 (Figure 2D) indicating the presence of DKK3 in axonal spheroids. These results indeed suggest that DKK3 is present in abnormally enlarged vesicles in dystrophic neurites around Aβ plaques. This could affect the axonal transport of DKK3. Given that proteins present in dystrophic neurites have been correlated with defects in bidirectional transport in the axon (Stokin GB et al., 2005, PMID: 15731448; Sadleir KR et al., 2016, PMID: 26993139), both DKK3 turnover and secretion could be affected.

3. Why does only LTD induce DKK3 release? Why not general activation of neuronal activity? It would be important to test the relationship between DKK3 secretion and neuronal activity with optogenetics and chemogenetics.

We tested whether neuronal activity triggered increased extracellular DKK3 levels by subjecting neurons to chemical long-term potentiation (cLTP) or long-term depression (cLTD). However, only cLTD increased extracellular DKK3, which we then confirmed in brain slices (Figure S3, current Figure 2-figure supplement 3). This finding is not unexpected as it is well described that different patterns of activity can lead to different molecular outcomes. For example, high-frequency stimulation (HFS; an activity pattern that resembles LTP) and low-frequency stimulation (LFS; a different activity pattern resembling LTD) leads to opposing effects on surface levels of the Wnt receptor Frizzled-5 (Fz5) (Sahores M et al., 2010, PMID: 20530549). Furthermore, cLTP increases Fz5 s-acylation, an important post-translational modification that regulates the surface levels of Fz5, whereas cLTD decreases it (Teo S et al., 2023, PMID: 37557176). Another example is the BDNF receptor TrkB. Surface TrkB is increased by tetanic stimulation, which also induces LTP as HFS or cLTP, but not by LFS (Du J et al., 2000, PMID: 10995446). Our findings suggest that DKK3 might contribute to synaptic changes underlying cLTD. Future experiments using chemogenetics or optogenetics might elucidate the role of DKK3 in activity-induced synaptic changes.

4. Are Abeta oligomer treatment-dependent increases in DKK3 protein levels in the cellular lysate and the extracellular fraction also suppressed by APV?

Our results in AD mice indicate that increased DKK3 release is dependent on NMDAR activation. To investigate if amyloid-β oligomers (Aβo) increase DKK3 levels in the cell lysate and extracellular fractions through NMDAR, we blocked these receptors in hippocampal neurons using AP-V (Figure S2G, H, current Figure 2 figure supplement 2 G, H). In these experiments, we use a lower concentration of Aβo (200nM of Aβ1-42) to avoid any potential cytotoxic effect. In line with our previous results using a higher concentration of Aβo, we observed that Aβo markedly increased DKK3 levels both in the cell lysate and in the extracellular fraction compared to the reverse Aβ42-1 control peptide. Kruskal-Wallis with Dunn’s test showed a trend to a reduced levels of DKK3 in the extracellular fraction when we compared neurons treated with Aβo and APV with those neurons treated with Aβ and vehicle (p = 0.0726). However, this reduced levels of DKK3 in the extracellular fraction reached statistical significance using a t-test (p = 0.0384). No differences were observed between the reverse control peptide and Aβo and APV conditions. These results suggest that blockade of the NMDAR partially occludes the ability of Aβo to increase DKK3 levels in the extracellular fraction.

5. Why does DKK3 shRNA only downregulate inhibitory synapses but not excitatory synapses in the WT brain slice? Does this mean that in the WT brain, other DKK proteins (without changes in their expression as shown in Fig S6) are sufficiently expressed and compensate for the roles of DKK3 in excitatory synapse integrity?

The reviewer points out an interesting result. In J20 mice, DKK3 knockdown affects both excitatory and inhibitory synapse density (Figure 6B, C). In Figure 3B, D, we show that in vivo downregulation of DKK3 leads to an increased number of inhibitory synapses without affecting excitatory ones in the brain of WT animals. These results indicate that in a healthy brain (WT), DKK3 is required for the maintenance of inhibitory synapses but not for excitatory synapses under our experimental conditions. Furthermore, DKK3 partially shares the mechanism of action with DKK1 as both DKK proteins promote excitatory synapse loss through the Wnt/GSK3β pathway (Figure 4A-C) (Marzo A et al., 2016, PMID: 27593374). Therefore, it is possible that endogenous DKK1 levels in the hippocampus could compensate for the reduced expression of DKK3 resulting in the lack of changes in excitatory synapse number when DKK3 is knockdown in WT animals.

6. Manipulating DKK3 in WT brains only affects Gephyrin but not VGAT, but in J20, both Gephyrin and VGAT seem to be affected by DKK3 shRNA (Fig 6). The authors need to provide the pre vs post synapse number in Fig 6 and discuss the potential differences.

We have now included the quantification of excitatory and inhibitory pre- and postsynaptic puncta for 4-months old (Figure S6B, C, current Figure 6-figure supplement 1, B, C) and 9-months old (Figure S6D, E, current Figure 6-figure supplement 1, D, E) WT and J20 mice. At 4-months old, the density of Homer1 puncta for excitatory synapses and both vGAT and Gephyrin for inhibitory synapses was increased and decreased respectively by knocking down DKK3 in the J20 mice. At 9-months, strong trends were observed in all the synaptic markers when downregulating DKK3, but significance was only reached for Homer1 puncta.

7. Where are the Wnt receptors expressed? Are they exclusively expressed in neurons? Can the authors exclude the potential involvement of glial cells in this process?

In neurons, Wnt receptors can be expressed in the synaptic terminals. For example, Wnt receptor Frizzled-5 is located at the presynaptic terminal and the dendritic shaft but not at spines (Sahores M et al., 2010, PMID: 20530549; McLeod F et al., 2018, PMID: 29694885), whereas Frizzled-7 is located at the dendritic shaft and spines (McLeod F et al., 2018, PMID: 29694885). In addition, the Wnt co-receptor LRP6 is present at both pre- and postsynaptic sites in excitatory synapses (Jones ME et al., 2023, PMID: 36638182). Kremen1, another receptor for Dkk proteins, is also highly expressed in the brain and our unpublished superresolution results show that this receptor is present in both pre- and postsynaptic sites of 53% of excitatory and 30% of inhibitory synapses. However, these receptors are not exclusively expressed in neurons and many of them are also highly expressed in astrocytes (Zhang Y et al., 2016, PMID: 25186741). Based on the literature and our findings, we cannot rule out the possibility that DKK3 may signal to other cell types such as astrocytes, which could also contribute to changes in synapse density. However, recombinant DKK3 induces structural and functional changes in excitatory and inhibitory synapses within 3-4h (Figure 3), suggesting that DKK3 acts on neurons leading to synaptic changes.

8. Does the shRNA treatment of DKK3 affect the size and number of amyloid plaques in the AD mice?

We thank the reviewer for raising this very important question. We have now evaluated the impact of DKK3 knockdown in Aβ pathology in the J20 mice. We did not observe differences in the Aβ coverage nor the averaged number and size of Aβ plaques when DKK3 was silenced in the CA3 (Figure S6F, current Figure 6-figure supplement 1, F). Therefore, the changes we observe in excitatory and inhibitory synapse density around plaques after knocking down DKK3 are unlikely to be due to changes in Aβ plaques.